# HybridVLA: Collaborative Autoregression and Diffusion in a Unified Vision-Language-Action Model

## Abstract

A fundamental objective of manipulation policy design is to endow robots to comprehend human instructions, reason about scene cues, and execute generalized actions in dynamic environments. Recent autoregressive vision-language-action (VLA) methods inherit common-sense reasoning capabilities from vision-language models (VLMs) for next action-token prediction. However, these methods quantize actions into discrete bins, which disrupts the continuity required for precise control. In contrast, existing diffusion-based VLA methods incorporate an additional diffusion head to predict continuous actions solely conditioned on feature representations extracted by the VLM, without fully leveraging the VLM's pretrained reasoning capabilities through token-level generation. To address these limitations, we introduce HybridVLA, a unified framework that absorbs the continuous nature of diffusion-based actions and the contextual reasoning of autoregression within a single large language model. To mitigate interference between the two generation paradigms, we propose a collaborative training recipe that seamlessly incorporates diffusion denoising into the next-token prediction process. With this recipe, we find these two action prediction methods not only reinforce each other but also exhibit varying strength across different tasks. Therefore, we design a collaborative action ensemble mechanism that adaptively fuses both predictions, leading to more robust control. HybridVLA outperforms previous state-of-the-art VLA methods by 14% and 19% in mean success rate on simulation and real-world tasks, respectively, while demonstrating stable manipulation in unseen configurations.

## 1 Introduction

Developing human-like robots capable of performing manipulation tasks demands intelligent policies [1, 2, 3]. In dynamic and unstructured real-world environments, such policies need to interpret human instructions and generalize across a wide range of complex tasks [4]. Recently, vision-language models (VLMs) [5, 6, 7, 8] have brought forth dramatic breakthroughs in instruction following and common-sense reasoning, driven by pretraining on internet-scale image-text pairs. Building on this success, several studies have extended VLMs into vision-language-action (VLA) models, enabling them to predict low-level action poses for robotic manipulation [9, 10, 11]. This paradigm outlines a promising roadmap for building foundation models to facilitate generalist robots.

On the one hand, autoregressive VLA methods [9, 11, 10, 15] emulate the reasoning paradigm of VLMs for next token prediction, effectively leveraging their large-scale pretrained knowledge and reasoning capabilities. While such methods enable generalized manipulation skills [10], they quantize continuous actions into discrete bins by adding new embeddings into the vocabulary in large language models (LLMs), which disrupts the continuity of action pose and hinders precise control [16]. On the other hand, building on the success of diffusion models in content generation [17, 18, 19, 20],

Submitted to 39th Conference on Neural Information Processing Systems (NeurIPS 2025). Do not distribute.

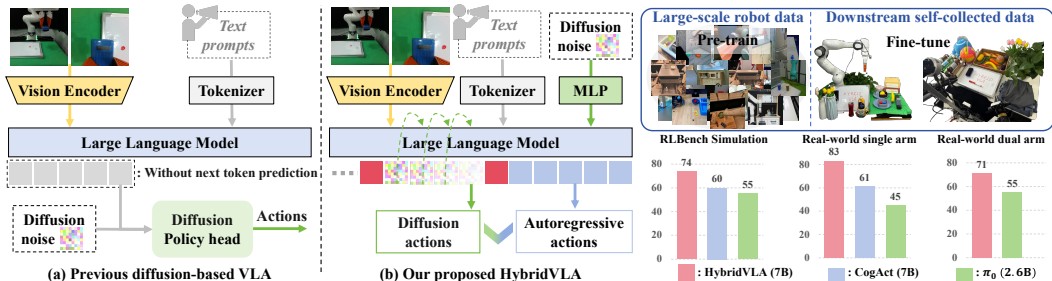

Figure 1: **(a)** Unlike recent diffusion-based VLA methods [12, 13, 14] that attach a separate diffusion head after VLMs, **(b)** HybridVLA innovatively integrates diffusion and autoregressive action prediction within a single LLM, fully leveraging the continuity of diffusion and the reasoning capabilities of autoregressive modeling. It is pretrained on large, diverse, cross-embodiment real-world robot datasets and further fine-tuned on downstream, self-collected data. HybridVLA achieves remarkable performance across various tasks involving both single-arm and dual-arm robots.

diffusion policies have been introduced in robotic imitation learning [21, 22, 23, 24, 25, 26]. Recent diffusion-based VLA methods [13, 14, 16, 12] incorporate a diffusion head after the VLM, leveraging probabilistic noise-denoising for action prediction. While these methods enable precise manipulation, the diffusion head operates independently of the VLM and lacks internet-scale pretraining. Moreover, since the head relies solely on VLM-extracted feature representations as input conditions, these methods fail to fully leverage the VLM's pretrained reasoning capabilities through next-token prediction. Given these advantages and limitations, a question arises: "*How can we elegantly construct a unified VLA model that seamlessly integrates the strengths of both autoregressive and diffusion policies, rather than simply concatenating them?*"

To this end, we propose HybridVLA, a unified framework that equips VLMs with both diffusion and autoregressive action prediction capabilities, enabling mutual reinforcement between them to facilitate robust execution across diverse tasks. As shown in Figure 1, unlike previous diffusion-based VLA methods [13, 14] that append an independent diffusion head after the LLM (Figure 1 (a)), we introduce a collaborative training recipe that seamlessly integrates diffusion denoising into the autoregressive next-token prediction process within a single LLM backbone (Figure 1 (b)). Specifically, since the token representations of discrete autoregressive tokens and continuous diffusion latents are inconsistent, a token sequence formulation is designed to systematically organize multimodal inputs, diffusion tokens, and autoregressive tokens, which are linked through specialized marker tokens. Under our proposed collaborative optimization, as both generation methods share the LLM backbone, HybridVLA explicitly captures the continuous action representations from diffusion modeling along with the pretrained semantic reasoning of autoregressive generation, allowing the two paradigms to reinforce each other. Meanwhile, we observe that diffusion generation excels in intricate tasks, while autoregression performs better in tasks requiring rich semantic understanding. Therefore, a collaborative action ensemble mechanism is proposed, where the two predictions are adaptively fused based on autoregressive action token confidence, improving robustness in manipulation.

To enhance generalization capability, we initialize HybridVLA with an internet-scale pretrained VLM [27], and design a step-by-step training approach [13, 10]. As shown in Figure 1, our model undergoes further pretraining on large, diverse, cross-embodiment robotic datasets, including Open X-Embodiment [28], DROID [29], and ROBOMIND [30], covering 760K trajectories and over 10K A800 GPU training hours. Subsequently, HybridVLA is fine-tuned on self-collected simulation data [31] and real-world data, achieving state-of-the-art (SOTA) manipulation performance across a variety of tasks with both single-arm and dual-arm robots. Meanwhile, HybridVLA demonstrates sufficient generalization capabilities to unseen manipulated objects, backgrounds, spatial positions, and lighting conditions during real-world testing, highlighting the effectiveness of our collaborative model design and training recipe. To optimize inference speed, we also introduce the HybridVLA-dif (7B) variant, which integrates diffusion and autoregressive generation during training but relies exclusively on diffusion-based actions for inference at 9.4 Hz. Our contributions are as follows:

- We propose HybridVLA, a unified model that seamlessly integrates diffusion and autoregressive action generation within a single LLM, effectively absorbing the continuous nature of diffusion-based actions and the contextual reasoning of autoregressive generation, thereby enabling mutual reinforcement and improving manipulation robustness.

- We introduce a collaborative training recipe that bridges the gap between the two action generation approaches, enabling mutual reinforcement through a shared LLM backbone. Additionally, we propose a collaborative action ensemble mechanism that adaptively fuses diffusion- and autoregressive-based actions, enhancing manipulation robustness.
- Our proposed method achieves SOTA performance across diverse tasks while demonstrating strong generalization to several unseen configurations.

## 2    Related Work

Traditional robotic manipulation primarily relies on state-based reinforcement learning [32, 33, 34, 35], whereas recent approaches [36, 37, 38, 21] integrate visual observations for imitation learning. Building on the strong reasoning capabilities of vision-language models (VLMs) [5, 6, 7, 39], recent research has integrated them into robotic manipulation [40, 41, 42, 43].

**Vision-language-action (VLA) models.** Some studies [2, 1, 3, 44] enable robots to interpret both language and visual observations, automatically generating task plans. Meanwhile, vision-language-action (VLA) models leverage the inherent reasoning abilities of VLMs to predict low-level SE(3) poses. Specifically, RT2[9] quantizes 7-DoF actions into discrete bins for autoregressive pose prediction. Building on this, ManipLLM[11] incorporates affordance priors through chain-of-thought reasoning, while OpenVLA[10] performs large-scale pretraining on the Open X-Embodiment dataset[28]. FAST [15] applies the discrete cosine transform to enable fast and scalable training of autoregressive-based VLA models. To support continuous action prediction, some VLA approaches [45, 46, 47, 48] incorporate a policy head, such as an MLP or LSTM [49], and use regression loss for imitation learning. However, quantization in autoregressive methods disrupts action continuity, while regressive methods fail to incorporate probabilistic action representations.

**Diffusion models in robotics.** Building on the success of diffusion models in content generation [17, 18, 19, 20], diffusion policies have been applied in robotics, including reinforcement learning [50, 51], imitation learning [21, 52, 53, 25, 26], grasping [54, 55, 56], and motion planning [57, 58]. Following this, 3D Diffusion Actor [23] and DP3 [21] employ diffusion models to interpret point cloud data. Octo [59] and RDT-1B [60] augment a transformer with a diffusion head to predict flexible actions.

**Diffusion-based VLA models.** To integrate diffusion with VLMs, $\pi_0$ [13] adds a diffusion expert head that generates actions through flow matching, while TinyVLA [61] incorporates a simple diffusion head after the lightweight VLM. CogACT [14] and DiVLA [16] decouple reasoning and action prediction into the VLM and an injected diffusion head, respectively. Following this architecture, some works [12, 62, 63] introduce a dual-system design to enable control at different frequencies. However, in these methods, the diffusion head operates as a separate module and treats the VLM as a multimodal feature extractor, limiting its ability to fully exploit pretrained reasoning capabilities through next-token prediction. In general scenarios, some works [64, 65, 66, 67] jointly tackle multimodal understanding and generation, while others [68, 69, 70] integrate diffusion into autoregressive transformers. Unlike prior methods focused on image and language generation quality, HybridVLA introduces a robotics-specific collaborative training strategy that integrates diffusion action generation into next-token prediction within a single LLM, enabling mutual enhancement.

## 3    HybridVLA Method

**Overview.** Existing diffusion-based VLA methods [13, 16, 14] append a separate diffusion head after the VLM. However, these methods overlook the LLM's core contextual reasoning mechanism (next-token prediction) acquired through internet-scale pretraining, since the head relies solely on VLM-extracted multimodal features from a single forward pass as diffusion conditions. In contrast, HybridVLA injects diffusion denoising into the next-token prediction process, equipping a single LLM with both diffusion and autoregressive action generation capabilities. To construct HybridVLA, we first describe the model architecture in Section 3.1. Since simply merging the two generation methods could cause inconsistency, we introduce a collaborative training recipe in Section 3.2. To further enhance robustness, we propose a collaborative action ensemble mechanism in Section 3.3.

**Problem Statement.** At time t, each demonstration consists of image observations $o_t$, language description $l_t$, and the current robot state $r_t$. Our model $\pi$ aims to predict action $a$ to control the robot arms, which can be formulated as: $\pi : (o_t, l_t, r_t) \rightarrow a_{t+1}$. Following [10, 14], the

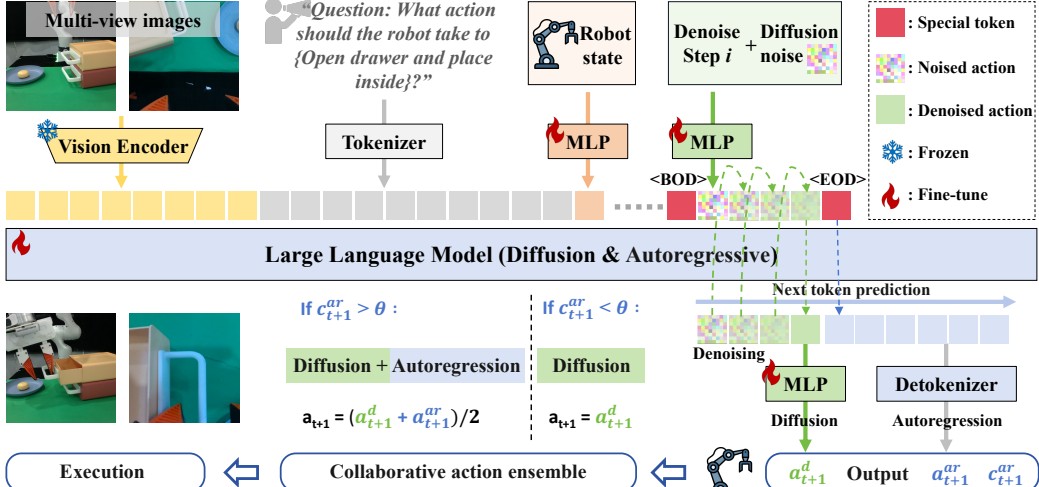

Figure 2: **HybridVLA Framework.** All multimodal inputs are encoded into tokens and subsequently organized into our designed token sequence formulation within the LLM's embedding space. For diffusion tokens, HybridVLA simultaneously projects the denoising timestep and noise into continuous vector representations. During inference, we adopt DDIM [71] with four sampling steps, where the corresponding noisy samples are iteratively fed into the LLM to predict the noise at each step. The marker tokens, <BOD> (Beginning of Diffusion) and <EOD> (End of Diffusion), are introduced to bridge the two generation paradigms. Subsequently, autoregressive actions are generated via standard next action-token prediction, explicitly conditioned on the preceding tokens. Our collaborative training recipe integrates knowledge from both generation paradigms into the shared LLM, enabling them to reinforce each other and be adaptively ensembled for robot arm control.

action $a$ represents the end-effector pose, which uses 7-DOF and 14-DOF for single-arm and dual-arm control, respectively. Each 7-DOF action includes 3-DOF for relative translation offsets ($[\Delta x, \Delta y, \Delta z] \in \mathbb{R}^3$), 3-DOF for rotation (Euler angles $\in \mathbb{R}^3$), and 1-DOF for the gripper state (open/closed $\in \mathbb{R}^1$). The ground truth (GT) and the model-predicted action are in SE(3), formulated as: $a = [\Delta x, \Delta y, \Delta z, Roll, Pitch, Yaw, 0/1]$.

## 3.1 HybridVLA Architecture

**Pretrained VLM base.** This section presents the architecture and workflow of HybridVLA, available in two model sizes, using 7B and 2.7B large language models (LLMs). Following [10], both HybridVLA(7B) and HybridVLA(2.7B) inherit the base architecture from Prismatic VLMs [27], initializing with the corresponding large-scale pretrained VLM parameters. We first introduce the two basic components, vision encoders and the LLM, as shown in Figure 2.

**Vision encoders.** HybridVLA leverages powerful vision encoder combinations, such as DINOv2 [72] and SigLIP [73], to capture rich semantic features $f_d \in \mathbb{R}^{B \times N_v \times 1024}$ and $f_s \in \mathbb{R}^{B \times N_v \times 1152}$. $B$ and $N$ represent batch size and token sequence length, respectively. These features are concatenated along the channel dimension to form $f_v \in \mathbb{R}^{B \times N_v \times 2176}$, which is subsequently projected into the LLM's word embedding via a projection layer. HybridVLA(2.7B) uses only the CLIP [74] model as its vision encoder. When processing multi-view images, a shared vision encoder extracts features, which are then concatenated along the token dimension.

**LLM.** HybridVLA adopts 7B LLAMA-2 [75] as LLM, responsible for multimodal understanding and reasoning. Language prompts are encoded into embedding space $f_l \in \mathbb{R}^{B \times N_l \times 4096}$ using the pre-trained tokenizer, then concatenated with visual tokens and input into LLM. The other specially designed LLM inputs (e.g., diffusion noise) are presented in the next section, and the output tokens are processed in two ways. First, diffusion-based action ($a_{t+1}^d$) generation through a denoising process, where an MLP maps the tokens into the action space. Second, autoregressive-based action generation ($a_{t+1}^{ar}$) is performed using a detokenizer [10], which also computes the mean confidence ($c_{t+1}^{ar}$) of the predicted tokens, serving as a guiding factor for the collaborative action ensemble. For HybridVLA (2.7B), the workflow remains the same as that of HybridVLA (7B) but utilizes the 2.7B Phi-2 [76] as the LLM. In the next section, we introduce how to simultaneously equip a single LLM with diffusion and autoregressive action generation capabilities.

## 3.2 Collaborative Training Recipe

Combining continuous diffusion and discrete autoregressive action generation within a single LLM presents challenges such as instability and inconsistency in the next-token prediction process. To address this, we propose a collaborative training recipe that includes a token sequence formulation, hybrid objectives, and structured training stages.

**Token sequence formulation design.** As shown in Figure 2, this design aims to organize multimodal tokens, such as robot state, diffusion noise, and autoregressive tokens, within the LLM's embedding space into a unified and ordered token sequence, enabling coordination between the two generation paradigms during the next-token prediction process. For the **robot state**, we integrate it into the LLM to enhance temporal consistency in action generation. Instead of discretizing the robot state and merging it with the language query [11] (Type 3 of Table 1), we employ a learnable MLP to map the robot state directly into the embedding space, $f_r \in \mathbb{R}^{B \times 1 \times 4096}$.

The motivation is that diffusion action tokens are generated using all preceding tokens as conditions. Introducing discrete robot states could negatively impact the diffusion prediction of continuous actions. For **diffusion-based actions**, we predict them through a diffusion denoising process. During training, the denoising step $i$ and noisy actions $a_t^i$ are projected into the LLM's word embeddings through an MLP, represented as continuous vectors. To seamlessly connect previous multimodal tokens, diffusion tokens, and subsequent discrete tokens within a sequence, we introduce special beginning-of-diffusion (<BOD>) and end-of-diffusion (<EOD>) tokens to encapsulate the diffusion tokens. This design not only clarifies the boundaries between diffusion and autoregressive generation but also prevents confusion in the next-token prediction process, such as

Table 1: Token sequence formulations. All models are trained with both generation methods. Dif and AR denote evaluations using only diffusion-generated or autoregressive-generated actions, respectively, across 10 RLBench tasks.

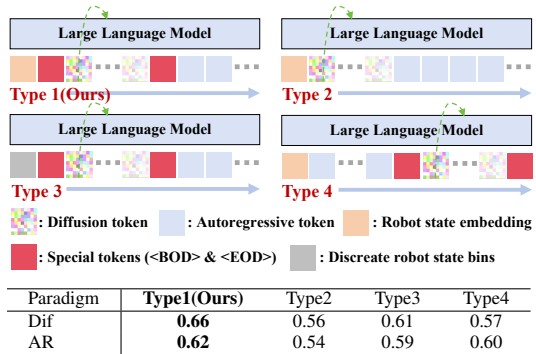

| Paradigm | Type1(Ours) | Type2 | Type3 | Type4 |
|---|---|---|---|---|
| Dif | **0.66** | 0.56 | 0.61 | 0.57 |
| AR | **0.62** | 0.54 | 0.59 | 0.60 |

avoiding diffusion tokens directly predicting masked discrete tokens (Type 2 of Table 1). For **autoregressive actions**, we quantize the end-effector pose into discrete bins and replace part of the vocabulary in the LLM [10], which is then tokenized into a sequence of discrete tokens. Due to the autoregressive nature of LLMs [77], both the question and the answer, including the discrete action ground truth (GT), are provided during training, whereas only the question is available during inference. Therefore, placing autoregression before the diffusion tokens may cause action GT leakage (Type 4 in Table 1), as all preceding tokens (which contain GT during training) serve as conditions in diffusion modeling. To avoid this, we position diffusion tokens before autoregression to explicitly provide continuous latent conditions for subsequent token prediction. Moreover, since diffusion operates on noise, it naturally circumvents the risk of information leakage.

**Hybrid objectives.** To simultaneously train diffusion and autoregressive action generation, we require two distinct loss functions. For the diffusion part, following previous diffusion policies [21], we minimize the mean squared error between the predicted noise ($\epsilon_\pi$) from the VLA model and the GT noise ($\epsilon$). The loss function is defined as follows: $L_{dif} = E_{a,i,c}||\epsilon - \epsilon_\pi(a_t^i, i, c)||^2$, where $\epsilon \sim N(0,1)$ and $c$ denote the condition. Additionally, classifier-free guidance [78] is not used in order to ensure stable robot arm behavior [60]. For the autoregressive part, the cross-entropy loss ($L_{ce}$) is adopted to supervise the discrete output. With our designed token sequence formulation, the two losses can be seamlessly combined for collaborative penalization, defined as: $L_{hybrid} = L_{dif} + L_{ce}$. Since $L_{dif}$ and $L_{ce}$ penalize a shared LLM backbone, their gradients are jointly backpropagated, allowing the model to effectively absorb both the continuous characteristics of diffusion-based actions and the semantic reasoning representations derived from autoregressive generation, thereby enabling mutual reinforcement between the two paradigms.

**Structured training stage.** After loading the pretrained VLM parameters, HybridVLA undergoes two training stages with hybrid objectives: large-scale pretraining on open-source robotic data and fine-tuning on self-collected data. During pretraining, we train HybridVLA for 5 epochs on 35 datasets [28, 29, 29]. The pretrain datasets contain 760k robot trajectories, comprising 33m frames.

Due to dataset differences, pretraining relies solely on single 2D observations, whereas fine-tuning relies on either single or multi-view observations, depending on the downstream task. The details of the pretraining dataset are provided in Appendix A.1.

### 3.3 Collaborative Action Ensemble

During inference, HybridVLA takes visual, language, and robot state inputs to concurrently generate actions via both diffusion and autoregressive methods, and ensembles them for execution.

**Autoregressive actions.** As shown in Figure 2, the autoregressive generation begins after the special token <EOD>. Unlike previous autoregressive VLA methods [10, 11], HybridVLA's autoregressive generation additionally conditions on continuous action representations derived from the latent features of diffusion tokens. This results in superior manipulation performance compared to independent autoregressive discrete generation paradigms that lack explicit continuous latent conditioning, as demonstrated in the ablation study.

**Diffusion actions.** When generating diffusion actions, we append the special token <BOD> after the previous condition tokens to indicate that the model should perform the denoising process. We employ DDIM [71] with $n$ sampling steps. In HybridVLA, we observe that the number of inference denoising steps can be reduced to 4 without causing any performance degradation. As illustrated in the denoising process of Figure 2, we repeat the process for 4 DDIM steps by feeding the noisy sample from the previous step into the LLM to predict the noise token for the current step, thereby fully leveraging the LLM's contextual reasoning capabilities. In this way, we effectively inherit the LLM's pretrained knowledge and seamlessly integrate diffusion generation into the next-token prediction process. Moreover, since we deliberately place the diffusion action tokens before the autoregressive tokens, the autoregressive predictions cannot be directly used as diffusion conditions. However, as discussed in the previous section, both generation methods share the same LLM backbone, which is jointly trained with hybrid objectives. As a result, the LLM is able to absorb the unique knowledge from each generation paradigm, thereby enhancing its overall representation. To accelerate the sampling process, we introduce the KV cache before the diffusion tokens, forwarding conditional information, the denoising timestep, and pure noise only during the initial sampling step. In subsequent steps, the cached keys and values from the first pass are reused, while only the timestep and noise are iteratively forwarded. This strategy eliminates redundant computations and improves inference speed.

**Ensembled actions.** After obtaining the two types of actions under our collaborative training recipe, we empirically observe two phenomena. 1) Different action types demonstrate varying performance across tasks. Diffusion-based predictions excel in precise manipulation tasks, such as *Phone on base* and *Close laptop lid*, while autoregressive predictions perform better in tasks requiring scene semantic reasoning, such as *Water plants* and *Frame off hanger*. 2) The confidence of autoregressive tokens serves as a reliable indicator of action quality. In over 80% of successfully completed test samples, the average confidence of autoregressive action tokens exceeds 0.96. Quantitative evaluations are provided in Appendix B.1 and B.2. Therefore, as shown in Figure 2, we use the mean confidence of autoregressive tokens ($c_{t+1}^{ar}$) to guide the action ensemble. If the confidence exceeds $\theta$ ($\theta = 0.96$), we consider the autoregressive action ($a_{t+1}^{ar}$) sufficiently accurate and perform an average operation with the diffusion action ($a_{t+1}^{d}$). Otherwise, we rely solely on the diffusion action to control the robot.

## 4 Experiment

In Section 4.1, we compare the manipulation ability and inference speed of HybridVLA with previous VLA methods in simulation environments. The effectiveness of each component is validated in Section 4.2 and Appendix B. In Section 4.3, we present both quantitative and qualitative manipulation results of HybridVLA in real-world scenarios, including single-arm and dual-arm robot tasks. The generalization capabilities of HybridVLA are examined in Section 4.4, testing on unseen manipulated instances, background, spatial positions, and lighting conditions.

### 4.1 Simulation Experiment

**Simulation benchmark.** To systematically evaluate, we select the RLBench [31] benchmark in the CoppeliaSim simulator, which contains 10 different tabletop tasks. These tasks, performed using a Franka Panda robot and a front-view camera, include *Close box*, *Close Laptop*, *Toilet seat down*,

Table 2: **Comparison of HybridVLA and baselines on RLBench.** We train all methods in the Multi-task setting [79] and report the success rates (S.R.). The success condition follows the definition in RLBench. (7B), (2.7B), and (2.6B) refer to the sizes of the LLM used in the VLA model.

| Models | Close box | Close laptop lid | Toilet seat down | Sweep to dustpan | Close fridge | Phone on base | Umbrella out | Frame off hanger | Wine at rack | Water plants | Mean S.R. & Var | Infer. speed |
|---|---|---|---|---|---|---|---|---|---|---|---|---|
| ManipLLM (7B) [11] | 0.50 | 0.80 | 0.40 | 0.20 | 0.80 | 0.35 | 0.10 | 0.25 | 0.15 | 0.20 | 0.38 ±0.042 | 2.2 Hz |
| OpenVLA (7B) [10] | 0.65 | 0.40 | 0.75 | 0.60 | 0.80 | 0.20 | 0.35 | 0.15 | 0.10 | 0.10 | 0.41 ±0.038 | 6.3 Hz |
| $\pi_0$ (2.6B) [13] | 0.90 | 0.60 | **1.00** | 0.30 | 0.90 | 0.25 | 0.35 | **0.75** | 0.05 | 0.45 | 0.55 ±0.035 | 13.8 Hz |
| CogACT (7B) [14] | 0.80 | 0.85 | 0.90 | 0.65 | 0.90 | **0.50** | **0.60** | 0.35 | 0.25 | 0.25 | 0.60 ±0.041 | 9.8 Hz |
| HybridVLA-dif (7B) | 0.85 | 0.75 | **1.00** | 0.80 | 0.95 | **0.50** | 0.50 | 0.30 | **0.70** | 0.25 | 0.66 ±0.040 | 9.4 Hz |
| HybridVLA (2.7B) | **1.00** | 0.80 | 0.90 | 0.80 | 0.90 | 0.25 | 0.20 | 0.45 | 0.25 | 0.25 | 0.58 ±0.031 | 12.3 Hz |
| HybridVLA (7B) | 0.85 | **0.95** | **1.00** | **0.90** | **1.00** | **0.50** | 0.50 | 0.70 | 0.50 | **0.50** | **0.74** ±0.037 | 6.1 Hz |

*Sweep to dustpan*, *Close fridge*, *Phone on base*, *Take umbrella out*, *Frame off hanger*, *Wine at rack*, and *Water plants*. The data are collected using pre-defined waypoints and the Open Motion Planning Library [80]. Following the frame-sampling method used in previous works [79, 81, 82], we construct the training dataset, with each task consisting of 100 trajectories.

**Training and Evaluation Details.** We compare our method with four previous SOTA VLA models, including autoregressive-based approaches such as ManipLLM [11] and OpenVLA [10], as well as diffusion-based methods like $\pi_0$ [13] and CogAct [14] with a DiT-base action head. Meanwhile, we categorize our method into three modes: HybridVLA (7B), HybridVLA (2.7B), and HybridVLA-dif (7B). All modes are jointly trained using our proposed collaborative training recipe; however, HybridVLA-dif relies solely on diffusion-based action generation during inference. To ensure a fair comparison, we load the official pretrained parameters provided by each method, adhering to their respective training settings. For HybridVLA, the single-view RGB input is resized to $224 \times 224$, and the robot state is consistent with predicted actions (7-DOF end-effector poses). During training, we use the AdamW optimizer with a fixed learning rate of 2e-5 to update both the LLM and the injected MLP parameters. Our models are trained for 300 epochs on 8 NVIDIA A800 GPUs with mixed-precision training. For evaluation, we follow [10, 14] and test all methods using 20 rollouts from the latest epoch checkpoint. Since RLBench employs a sampling-based motion planner [83], we evaluate each model three times per task and report the mean success rate along with its variance.

**Quantitative Results.** As shown in Table 2, HybridVLA(7B) achieves an average success rate of 74% across 10 distinct tasks, outperforming the previous SOTA autoregressive-based VLA (OpenVLA) and diffusion-based VLA (CogACT) by 33% and 14%, respectively. These results demonstrate that our method effectively combines the two generation approaches within a shared LLM backbone, simultaneously capturing the continuous characteristics of diffusion-based actions and the pretrained semantic reasoning capabilities learned through autoregression. Remarkably, compared to CogACT and $\pi_0$, HybridVLA-dif also achieves performance improvements of 6% and 11%, respectively. These results highlight that, unlike previous approaches which attach the diffusion head after the VLM, our method more effectively leverages the VLM's pretrained knowledge to fully unlock the potential of diffusion prediction. Finally, HybridVLA(2.7B) delivers satisfactory results, confirming our method's effectiveness in enhancing VLM manipulation capabilities across different model sizes.
**Inference Speed.** In Table 2, when tested on an NVIDIA 4090D GPU, HybridVLA-dif (7B) and HybridVLA (2.7B) achieve satisfactory control frequencies comparable to CogACT (7B) and $\pi_0$ (2.6B), thanks to the reduced DDIM denoising steps and the use of KV cache in HybridVLA. Note that all models are run with bfloat16 precision during inference, without employing action chunking.

## 4.2 Ablation Study

We conduct ablation experiments on 10 RLBench tasks, using the same training and evaluation settings as in the simulation experiments. **To evaluate the effectiveness of the Collaborative Training recipe (CTR),** we compare Ex1 with Ex2 and Ex3 with Ex4, as shown in Table 3. HybridVLA-dif (Ex1) and HybridVLA-ar (Ex3) are both trained under our proposed CTR that integrates diffusion and autoregressive action generation. Since diffusion tokens precede autoregressive tokens, HybridVLA-dif (Ex1) is evaluated solely on diffusion generation, while HybridVLA-ar (Ex3) performs diffusion denoising followed by autoregressive generation, but is tested only on autoregressive actions. Compared to Ex2 and Ex4, which are trained solely on individual generation methods, both HybridVLA-dif (Ex1) and HybridVLA-ar (Ex3) demonstrate improved manipulation performance. These results validate that our proposed CTR not only avoids negative interference between the two generation paradigms, but also effectively captures the continuous action representations from diffusion-based generation and the pretrained reasoning capabilities from autoregressive generation,

Table 3: **Impact of each component.** AR and Dif represent autoregressive and diffusion-based action generation, respectively. LSP denotes large-scale pretraining on assembled robotic datasets, while RSE refers to the injected robot state embedding. CTR and CAE represent our proposed collaborative training recipe with hybrid objectives and the collaborative action ensemble method.

|  | AR | Dif | LSP | RSE | CTR($L_{Hybrid}$) | CAE | **Mean↑** |
|---|---|---|---|---|---|---|---|
| Ex0 | ✓ | ✓ | ✓ | ✓ | ✓ | ✓ | 0.74 |
| Ex1 | - | ✓ | ✓ | ✓ | ✓ | - | 0.66 |
| Ex2 | - | ✓ | ✓ | ✓ | - | - | 0.60 |
| Ex3 | ✓ | - | ✓ | ✓ | ✓ | - | 0.62 |
| Ex4 | ✓ | - | ✓ | ✓ | - | - | 0.57 |
| Ex5 | ✓ | ✓ | - | ✓ | ✓ | ✓ | 0.22 |
| Ex6 | ✓ | ✓ | ✓ | - | ✓ | ✓ | 0.68 |

Table 4: **Generalization.** "Object", "Background", "Height", and "Lighting" denote unseen manipulated objects, backgrounds, spatial positions, and lighting conditions, respectively. The image above depicts the unseen test scenarios, with red boxes marking the key differences.

| Task | Pick and place(single arm) | | Lift ball and place(dual arm) | |
|---|---|---|---|---|
| Scenario | HybridVLA | Cogact | HybridVLA | $\pi_0$ |
| Original | 0.90 | 0.80 | 0.80 | 0.65 |
| Object | 0.60(-33%) | 0.45(-43%) | 0.75(-6%) | 0.60(-8%) |
| Background | 0.80(-11%) | 0.50(-37%) | 0.60(-25%) | 0.50(-23%) |
| Height | 0.75(-17%) | 0.50(-37%) | 0.60(-25%) | 0.45(-31%) |
| Lightning | 0.70(-22%) | 0.60(-25%) | 0.75(-6%) | 0.55(-15%) |

enabling mutual reinforcement. The various token formulation designs used in our training recipe are explored in Table 1 and Section 3.2. **For large-scale pretraining (LSP),** we compare Ex5 with Ex0. Although Ex5 is initialized with pretrained VLM parameters, it suffers from a significant drop in accuracy, highlighting the essential role of large-scale pretraining on robot datasets in ensuring stable control. **For robot state embedding (RSE),** by comparing Ex6 with Ex1, we observe that injecting robot state information enhances the model's temporal consistency during action prediction. Due to space limitations, Appendix B.2 provides additional ablation studies on: (1) confidence thresholds in the collaborative action ensemble, (2) the influence of the KV cache on inference speed, and (3) the impact of DDIM sampling steps on performance.

### 4.3 Real-World Experiment

**Self-collected Data.** For single-arm tasks, we use a Franka Research 3 robot with a static front-view and a wrist-view camera. We perform 5 tasks: 1) *Pick and place*, 2) *Unplug charger*, 3) *Open drawer and place inside*, 4) *Pour water*, 5) *Wipe blackboard*. For each task, 100 demonstrations are collected via teleoperation using a SpaceMouse device. For dual-arm tasks, we use an AgileX dual-arm robot equipped with a static exterior view, a right-wrist view, and a left-wrist view camera. We conduct 5 coordinated dual-arm tasks: 1) *Pick and place*, 2) *Lift ball and place*, 3) *place two bottles at rack*, 4) *Wipe blackboard*, 5) *Fold shorts*. Similarly, 100 demonstrations are collected for each task using master-puppet teleoperation. Additional details are provided in Appendix A.2.

**Training and Evaluation Details.** We evaluate HybridVLA (7B) and HybridVLA-dif (7B) against previous VLA methods, $\pi_0$ [13] and CogAct [14]. The implementation details remain consistent with our simulation experiments, except for using two-view inputs for single-arm tasks and three-view inputs for dual-arm tasks. For evaluation, we use the checkpoint from the latest epoch to perform 20 rollouts across diverse tabletop positions.

**Quantitative and Qualitative Results.** In Table 5, HybridVLA and HybridVLA-dif achieve outstanding performance across single-arm tasks. For *Pick and place* and *Unplug charger*, HybridVLA achieves success rates of 90% and 95%, respectively, demonstrating accurate object position prediction. For *Pour water*, HybridVLA and HybridVLA-dif outperform the previous SOTA method by 35% and 30%, respectively, showcasing their ability to comprehend object relationships and predict precise rotations. The superior performance on *Wipe blackboard* and *Open drawer and place inside* further underscores the robustness of our method in long-horizon tasks. For dual-arm tasks, we extend the action dimensions of both diffusion and autoregressive tokens to 14-DOF, representing the 7-DOF end-effector poses for both the right and left arms. Our method consistently outperforms previous VLA approaches across five distinct tasks, highlighting HybridVLA's ability to effectively leverage VLMs' reasoning capabilities for dual-arm coordination in complex scenarios. Furthermore, in the lower part of Table 5, we present visualizations of the manipulation processes performed by our method, which accurately predicts actions across various task demands, including precise positioning and rotation, dual-arm coordination, and scene understanding. Additional qualitative results and failure case analyses are provided in Appendix C and Appendix D, respectively, and execution videos are available in the supplementary materials.

Table 5: **Real-world experiments.** All methods are trained in a single-task setting [22], with success determined by human evaluation. Since CogAct lacks support for multi-view images, which are crucial for dual-arm tasks [13, 37], we conduct our dual-arm comparison solely with $\pi_0$.

| Models | Franka single-arm robot | | | | | | AgileX dual-arm robot | | | | | |
|---|---|---|---|---|---|---|---|---|---|---|---|---|
| | Pick and place | Unplug charger | Pour water | Wipe blackboard | Open drawer and place inside | Mean. S.R. ↑ | Pick and place | Lift ball and place | Place bottles at rack | Wipe blackboard | Fold shorts | Mean. S.R. ↑ |
| $\pi_0$ (2.6B) [13] | 0.50 | 0.35 | 0.45 | 0.35 | **0.60** | 0.45 | 0.75 | 0.65 | 0.40 | 0.30 | 0.65 | 0.55 |
| CogACT (7B) [14] | 0.80 | 0.70 | 0.40 | 0.65 | 0.50 | 0.61 | - | - | - | - | - | - |
| HybridVLA-dif(7B) | **0.85** | **0.95** | **0.75** | **0.85** | 0.60 | **0.80** | 0.80 | 0.75 | 0.60 | 0.45 | 0.70 | 0.66 |
| HybridVLA(7B) | **0.90** | **0.95** | **0.80** | **0.85** | **0.65** | **0.83** | **0.90** | **0.80** | 0.60 | **0.55** | 0.70 | **0.71** |

Single-arm real-world tasks — Dual-arm real-world tasks

## 4.4 Generalization Experiment

Since CogAct and $\pi_0$ excel in single-arm and dual-arm tasks, respectively, we design four common generalization experiments, comparing our HybridVLA with CogAct on the single-arm *Pick and place* task and with $\pi_0$ on the dual-arm *Lift ball and place* task. **1) Unseen manipulated objects.** In this scenario, we replace the training manipulated objects with a series of unseen objects, e.g., replacing the red block with a charger. As shown in the "Object" row of Table 4, our method demonstrates the smallest accuracy drop. These results indicate that, unlike previous diffusion-based VLA methods, HybridVLA effectively integrates diffusion into the autoregressive next-token prediction process, not only capturing the continuous characteristics of diffusion-based generation, but also preserving the object-level semantic reasoning capabilities of autoregressive generation. **2) Unseen background.** In this scenario, cluttered backgrounds are introduced during testing, such as adding unseen flowers around the manipulated object. HybridVLA still shows satisfactory results, further demonstrating that our collaborative training recipe effectively inherits the VLM's scene-level reasoning capabilities, enhancing robustness to environmental variations. **3) Unseen Spatial position.** Unlike position shifts within the same plane, we introduce height variations during testing, further challenging the model's spatial comprehension. As shown in the "Height" row of Table 4, HybridVLA consistently achieves precise manipulation even when encountering objects in previously unseen spatial positions. These results highlight that HybridVLA exhibits strong trajectory generalization capabilities through the ensemble of two action generation methods. **4) Unseen lighting conditions.** Finally, we introduce variations in lighting conditions, a common challenge in real-world environments. All methods maintain satisfactory performance, demonstrating that large-scale pretraining on robotic datasets enhances their generalization across diverse data distributions.

## 5 Conclusion and Limitation

In this paper, we introduce HybridVLA, a unified Vision-Language-Action (VLA) framework that equips a single LLM with both diffusion-based and autoregressive action generation capabilities. To bridge the gap between these two paradigms, we propose a collaborative training recipe that integrates diffusion denoising into the next-token prediction process, enabling mutual reinforcement and improving manipulation robustness. By effectively absorbing the continuous nature of diffusion-based action generation and the semantic reasoning capabilities of autoregressive methods, HybridVLA achieves outstanding performance and strong generalization across both simulation and real-world tasks. One limitation of HybridVLA is that its inference speed is constrained by the slower autoregressive generation, similar to prior autoregressive VLA methods [10, 9, 11]. However, our collaborative training enables mutual reinforcement between the two generation methods, allowing inference using only the diffusion process (HybridVLA-dif), achieving a 9.4 Hz inference speed. Finally, we state the broader impact of our work in Appendix E.

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

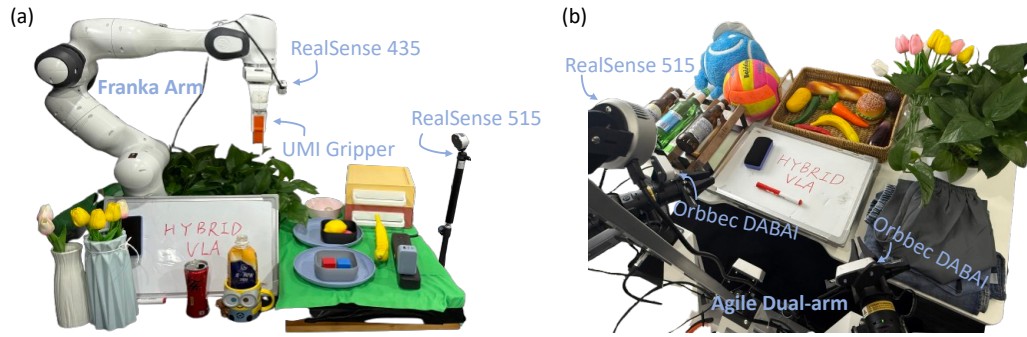

Figure 3: **Real-World Assets and Experimental Settings.** We provide visualizations of the assets used and the experimental settings for single-arm FR3 robot tasks and dual-arm AgileX robot tasks, respectively.

**Appendix A.** We begin by detailing the large-scale pretraining and self-collected real-world datasets.

**Appendix B.** Additional simulation experiments and ablation studies are presented.

**Appendix C.** We include further visualizations of both single-arm and dual-arm manipulation processes.

**Appendix E.** A brief conclusion and hope to our work's broader impact.

# A    Additional Dataset Details

## A.1    Large-scale Pretraining Dataset

Our pre-training dataset collection comprises 35 datasets, encompassing a total of 760k trajectories and 33m frames. Table 6 provides a comprehensive list of our pre-training datasets along with their respective sampling weights. The number of trajectories and the sampling weights can be automatically adjusted during dataset assembly. Following the prior data preprocessing approach [10], we reformulate the pre-training datasets to emphasize end-effector sequence control, ensuring alignment with the specific requirements of our model training. Due to inherent differences among datasets, only single 2D observations are used during pre-training. However, during fine-tuning, HybridVLA can accommodate both single- and multi-view observations depending on the task requirements. For instance, AgileX dual-arm robot tasks require three viewpoints—an ego view and two wrist camera views—to capture a comprehensive observation of the object while mitigating occlusions caused by the robot arm. HybridVLA processes multi-view images using a shared vision encode and then concatenates the visual feature along the token dimension. Notably, the difference in the number of images used during pre-training and fine-tuning does not impact manipulation performance in downstream tasks.

## A.2    Self-collected Real-world Dataset

The experimental assets and environments for the single-arm and dual-arm setups are shown in Figure 3 (a) and (b), respectively. For the single-arm setup, a 3D-printed UMI gripper [117] is attached to the Franka robot and is used across all baselines. We utilize RealSense 435 and RealSense 515 cameras to capture both wrist and front views. For the dual-arm setup, two Orbbec DABAI cameras are used to capture the left and right wrist views, while a RealSense 515 is mounted overhead to capture a static third-person view. We provide a detailed explanation of the real-world tasks and their success conditions. We begin by describing the single-arm tasks:

*1. Pick and place.* This task requires the robot to pick up a specifically colored block based on a language description and place it in a specifically colored bowl.

Table 6: The dataset name and sampling weight used in our mixed large-scale pretraining dataset.

| Training Dataset Mixture | |
| --- | --- |
| Fractal [38] | 9.1% |
| Kuka [84] | 27.8% |
| Bridge[85, 86] | 4.1% |
| Taco Play [87, 88] | 2.1% |
| Jaco Play [89] | 0.3% |
| Berkeley Cable Routing [90] | 0.2% |
| Roboturk [91] | 1.7% |
| Viola [92] | 0.7% |
| Berkeley Autolab UR5 [93] | 0.9% |
| Toto [94] | 1.5% |
| Language Table [95] | 3.1% |
| Stanford Hydra Dataset [96] | 3.2% |
| Austin Buds Dataset [97] | 0.2% |
| NYU Franka Play Dataset [98] | 0.6% |
| Furniture Bench Dataset [99] | 1.8% |
| UCSD Kitchen Dataset [100] | <0.1% |
| Austin Sailor Dataset [101] | 1.6% |
| Austin Sirius Dataset [102] | 1.2% |
| DLR EDAN Shared Control [103] | <0.1% |
| IAMLab CMU Pickup Insert [104] | 0.7% |
| UTAustin Mutex [105] | 1.6% |
| Berkeley Fanuc Manipulation [106] | 0.6% |
| CMU Stretch [107] | 0.1% |
| BC-Z [108] | 5.4% |
| FMB Dataset [109] | 5.0% |
| DobbE [110] | 1.0% |
| DROID [29] | 7.2% |
| Stanford Kuka Dataset [111] | 0.1% |
| Stanford Robocook Dataset [112] | 0.1% |
| Maniskill [113] | 6.3% |
| Berkeley RPT [114] | 0.1% |
| QUT Dexterous Manipulation [115] | 0.1% |
| RoboSet [116] | 1.8% |
| BridgeData V2 [86] | 4.7% |
| RoboMind [30] | 5.2% |

*2. Unplug charger.* The robot needs to grasp the charger at an optimal position and rotation, and then lift it to a certain height without slipping.

*3. Pour water.* The robot needs to first pick the bottle, then rotate it to a position slightly above the cup, and tilt it to perform the pouring action. The task is deemed successful only if the bottle opening is correctly aligned with the cup.

*4. Wipe blackboard.* The robot needs to first grasp an eraser and then use it to remove the red markings from a blackboard placed on the tabletop. The red markings are drawn on an unfixed region, and the task is considered successful only if they are completely erased.

*5. Open drawer and place inside.* The robot needs to open the top drawer, pick up the required objects based on the language description, place them in the opened drawer, and then close it. This task consists of four sequential sub-tasks: *open drawer*, *pick object*, *place object*, and *close drawer*. The task is considered complete once all sub-tasks have been successfully executed.

We then describe the details of dual-arm tasks:

*1. Pick and place.* The robot must use both its left and right arms to pick up two objects based on the language description and place them in the container.

*2. Lift ball and place.* Both the left and right arms must simultaneously make contact with the ball, which is secured between the two grippers. The arms coordinate their movements to transport the ball to the container while ensuring it does not slip. This task highly tests the model's dual-arm coordination capabilities.

*3. Place bottles at rack.* The left and right robot arms need to grasp the bottles placed on their respective sides and rotate them to position them parallel to the rack.

*4. Wipe blackboard.* Unlike the single-arm setting, the dual-arm setting requires one arm to hold the whiteboard while the other picks up the eraser and wipes off the red marker.

*5. Fold shorts:* This task requires folding a pair of shorts, involving two sequential steps. First, one pant leg is folded over the other to align them. Then, the pants are folded in half from top to bottom. Throughout the process, both arms must coordinate their movements. For example, in the first step, the left arm holds the bottom of the pant leg while the right arm grips the upper part, working together to complete the folding.

## B  Additional Quantitative Results

### B.1  Additional Simulation Experiments

In Table 7, we validate the first observed phenomenon mentioned in Section 3.3: different action types within our proposed framework exhibit varying performance across tasks. Meanwhile, we categorize our method into three modes: HybridVLA (7B), HybridVLA-ar (7B), and HybridVLA-dif (7B). All modes undergo joint training using our proposed collaborative training recipe; however, HybridVLA-ar and HybridVLA-dif rely exclusively on autoregressive-based and diffusion-based action generation during inference, respectively. The experiments are conducted in the RLBench simulator across 10 tasks, and evaluated based on success rate. Comparing HybridVLA-ar and HybridVLA-dif, HybridVLA-ar outperforms in 4 out of 10 tasks, while HybridVLA-dif leads in the remaining 6 tasks. These results validate our findings that, within the HybridVLA framework, diffusion-based predictions excel in precise manipulation tasks, such as *Phone on base*, *Toilet seat down*, and *Close laptop lid*, whereas autoregressive predictions perform better in tasks requiring scene-level semantic reasoning, such as *Sweep to dustpan*, *Water plants*, and *Frame off hanger*. Therefore, while collaborative training allows diffusion-based and autoregressive-based action generation to reinforce each other, assembling both methods results in more robust actions.

Table 7: **Detailed Simulation Experiments.** We validate that different action types within our proposed framework exhibit varying performance across tasks. All models undergo joint training using our proposed collaborative training recipe; however, HybridVLA-ar and HybridVLA-dif rely exclusively on autoregressive-based and diffusion-based action generation during inference, respectively. Underlining indicates the highest score between HybridVLA-ar and HybridVLA-dif.

| Models | Close box | Close laptop lid | Toilet seat down | Sweep to dustpan | Close fridge | Phone on base | Umbrella out | Frame off hanger | Wine at rack | Water plants | Mean. S.R. ↑ |
|---|---|---|---|---|---|---|---|---|---|---|---|
| HybridVLA-ar(7B) | **0.85** | 0.70 | 0.90 | 0.85 | 0.95 | 0.30 | 0.25 | 0.40 | 0.45 | **0.50** | 0.62 |
| HybridVLA-dif(7B) | **0.85** | 0.75 | 1.0 | 0.80 | 0.95 | 0.50 | 0.50 | 0.30 | 0.70 | 0.25 | 0.66 |
| HybridVLA(7B) | 0.85 | 0.95 | 1.0 | 0.90 | 1.0 | 0.50 | 0.50 | 0.70 | 0.50 | 0.50 | 0.74 |

Table 8: **Ablation Study.** We explore the impact of different confidence thresholds on the performance of ensemble actions.

| Threshold | 0.90 | 0.92 | 0.94 | 0.96 | 0.98 |
|---|---|---|---|---|---|
| Success rate | 0.66 | 0.64 | 0.70 | 0.74 | 0.69 |

### B.2  Additional Ablation Study

**The impact of confidence threshold in collaborative action ensemble.** The proposed collaborative ensemble strategy determines whether to use the action predicted by diffusion alone or the averaged output of both diffusion and autoregressive methods, guided by a mean confidence threshold derived from the autoregressive action token. In this experiment, we investigate the optimal confidence threshold required to ensure the accuracy of autoregressive actions and enhance the overall precision

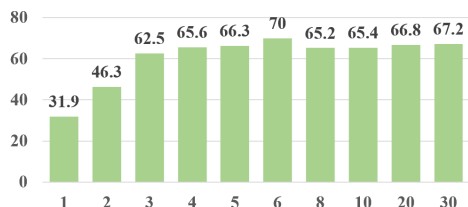

Figure 4: **The impact of denoising steps**, where the x-axis and y-axis represent the denoising steps and manipulation success rate.

of the ensemble-generated action. Specifically, as shown in Table 8, we vary the threshold from 0.90 to 0.98. We find that when the confidence threshold drops below 0.94, autoregressive predictions become unreliable, leading to a slight degradation in the performance of the ensemble action. Conversely, when the threshold reaches 0.98, the number of valid autoregressive actions becomes too limited, causing the performance of the ensemble action to closely match that of the diffusion-predicted action. Empirically, we conclude that setting the threshold to 0.96 ensures a stable action ensemble.

**The impact of KV cache in inference speed.** As described in Section 3.3, we adopt the KV cache to eliminate redundant computations and improve inference speed. In this experiment, we examine the extent to which this mechanism accelerates inference. With the KV cache enabled (Table 2 of the main paper), HybridVLA-dif achieves an average success rate of 66% across 10 simulation tasks with an inference speed of 9.4 Hz. Removing it results in a similar average success rate but reduces the inference speed to 5.0 Hz. Although the KV cache has typically been used in previous autoregressive VLA methods [10, 11], we are the first to integrate it into an LLM's diffusion-based action generation.

**The impact of denoising steps.** In Figure 4, we explore the relationship between manipulation performance and different denoising steps on HybridVLA-dif. Consistent with the findings of previous work [12, 60], we reduced the number of DDIM denoising steps of inference from 30 to 4 without observing a significant degradation in manipulation performance. To balance inference speed and accuracy, we set the diffusion denoising steps to 4 in our final implementation.

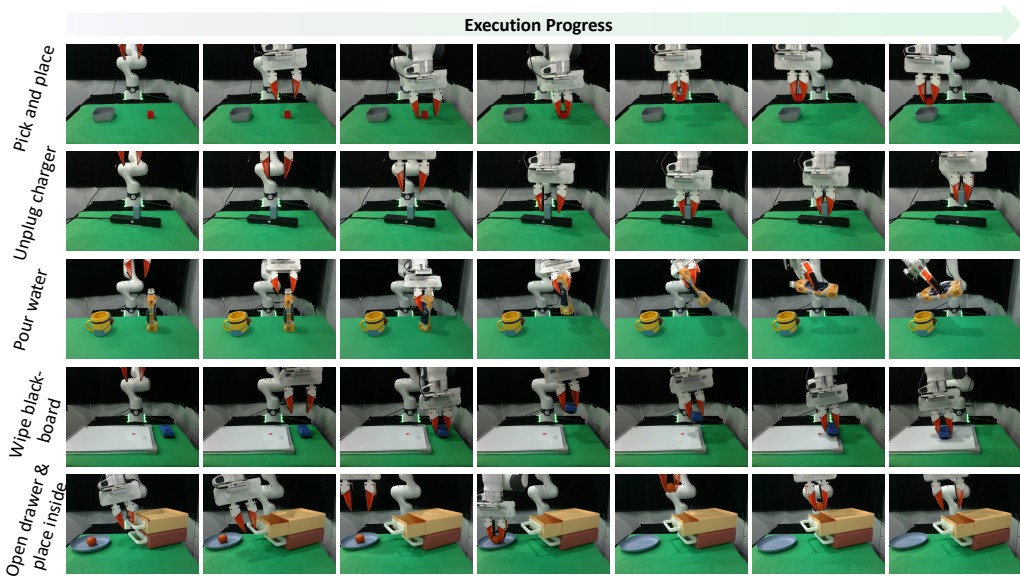

Figure 5: **Single-arm Execution Visualization**. We visualize key frames of the agent's execution process from the front perspective.

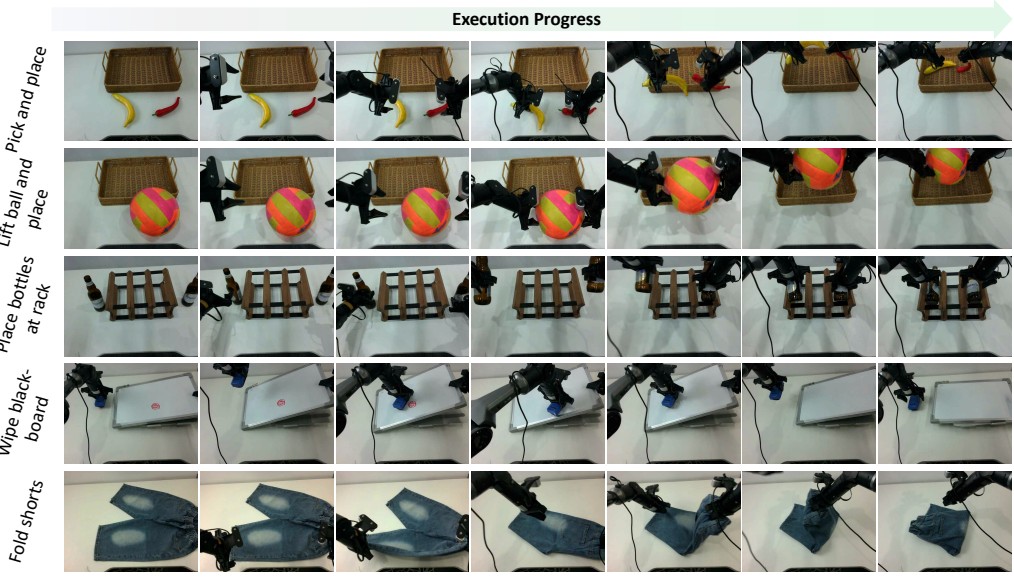

Figure 6: **Dual-arm Execution Visualization**. We visualize key frames of the agent's execution process from a static exterior view.

## C   Additional Visualizations

Figure 5 and Figure 6 illustrate keyframes of single-arm and dual-arm real-world execution processes. Notably, our Franka Research 3 (FR3) operates with controller version 5.6.0, libfranka version 0.13.3, Franka ROS version 0.10.0, and Ubuntu 20.04 with ROS Noetic. Under these software settings, the FR3 remains in *green* light execution mode with the FCI switch set to 'on'.

These tasks demonstrate HybridVLA's capability in accurately predicting position and rotation, as well as determining the precise timing for changing the gripper's open state. Additionally, the dual-arm tasks highlight HybridVLA's ability to coordinate both robotic arms, enabling it to complete tasks beyond the capability of a single arm, such as transporting a ball to a container. Notably, the single-arm task 'open drawer and place' and the dual-arm tasks 'wipe whiteboard' and 'fold shorts' are long-horizon tasks that involve at least three multi-step actions. These results further confirm that HybridVLA can reliably predict sequential actions, demonstrating the capability to complete long-horizon tasks.

## D   Failure Case Analysis.

Through extensive real-world experiments, we identify three primary failure categories that impact the performance of HybridVLA. The first category, **rotational prediction deviations**, is particularly evident in tasks requiring precise rotation control, such as *Pour water* and *Place bottle at rack*. These failures include accumulated errors in multi-step rotational movements and incorrect rotation angles when interacting with target objects. The second category pertains to pose predictions that exceed the robot's **degree of freedom limits**. The model sometimes predicts poses beyond the mechanical constraints of the Fr3 arm or AgileX dual-arm robot, generates target positions that fall outside the workspace boundaries, or produces kinematically infeasible configurations during complex transitions. The third category involves failures in **dual-arm coordination**, where both arms must collaborate to complete a task. Since the model predicts each arm's actions based on the current object state, any interaction by one arm can alter the object's state, potentially invalidating the previously predicted action of the other arm.

# E   Broader Impact

Our work proposed a collaborative framework to combine the continuous nature of diffusion-based action and the contextual reasoning of autoregression within a single LLM. This work focused on the innovation of the above VLA structure and does not have a direct impact on society. And we hope that this effort can promote the progress in the field of robot manipulation and open up a new paradigm for better providing foundation models in the field of embodiment intelligence, so as to promote the healthy, controllable and sustainable development of the entire field.

