# OpenReview forum: "HybridVLA: Collaborative Autoregression and Diffusion in a Unified Vision-Language-Action Model"
_NeurIPS.cc/2025/Conference — Submitted to NeurIPS 2025_

### Official Review · Reviewer_emK6 · 2025-06-30

**Clarity:** 3
**Significance:** 1
**Originality:** 3
**Rating:** 2
**Confidence:** 5

**Summary:**

HybridVLA introduces a novel unified Vision-Language-Action (VLA) framework. This framework integrates the diffusion models for continuous action and autoregressive models for discrete action within a single large language model (LLM). It proposes a collaborative training recipe that seamlessly incorporates diffusion denoising into the next-token prediction process. HybridVLA also designs a collaborative action ensemble mechanism that adaptively fuses both predictions based on autoregressive action token confidence. In simulation and real-world tasks, HybridVLA outperforms previous state-of-the-art VLA methods by 14% and 19% in mean success rate, respectively. It also demonstrates strong generalization capabilities.

**Questions:**

1. Why is the inference speed of HybridVLA-7B (6.3hz) faster than OpenVLA-7B (6.1hz)? HybridVLA need do at least 4 more denoising steps, which is equivalent to performing 4 more forward passes. If it's because HybridVLA has KV-cache while OpenVLA does not, I don't think this is fair.
2. Why is there no evaluation on the libero/SimplerEnv benchmark?

**Ethical Concerns:**

["NO or VERY MINOR ethics concerns only"]

**Final Justification:**

Since my concerns have not been fully addressed, I maintain my original score.

**Limitations:**

Lack of the fundamental mathematical analysis.

**Quality:**

2

**Strengths And Weaknesses:**

**Strengths**

1. HybridVLA outperforms previous state-of-the-art VLA methods in simulation (i.e. RLBench) and real-world tasks, respectively.


**Weaknesses**

1. Imprecise and Misleading Attribution of "Common-sense Reasoning":

The paper frequently attributes "common-sense reasoning" specifically to "autoregressive generation" and "next-token prediction". It contrasts this with existing diffusion-based VLA methods, claiming they "fail to fully leverage the VLM's pretrained reasoning capabilities through next-token prediction". This portrayal creates a potentially misleading dichotomy regarding the nature of LLM capabilities.
- LLM's Intrinsic Reasoning Prowess: The LLM's capacity for "common-sense reasoning" and "contextual understanding" is an inherent capability derived from its architecture and extensive pre-training on internet-scale text and image pairs. This powerful reasoning ability is a property of the LLM as a general VLM, and it is not exclusively dependent on, or solely manifested through, autoregressive "next-token prediction." An LLM's internal representations can encode rich semantic and contextual information regardless of the specific downstream generative task (discrete token prediction or continuous data generation).
- Diffusion Models' Utilization of Context: Diffusion models, when conditioned by features from a VLM, do leverage the VLM's contextual understanding. The VLM processes the multimodal input (images, language prompts) and generates sophisticated feature representations that implicitly contain the necessary contextual and semantic information. The diffusion model then utilizes these contextual features to guide its continuous generation process. The success of large-scale text-to-image diffusion models, which produce high-fidelity images conditioned solely on text embeddings (derived from LLM-like encoders), clearly demonstrates that diffusion can effectively utilize complex contextual reasoning without an explicit "next-token prediction" mechanism for the output itself. The paper's claim that diffusion needs to be "injected into next-token prediction" to fully leverage reasoning might overstate the unique role of autoregression in accessing the LLM's inherent contextual understanding.

2. Theoretical Ambiguity of "Unification" and Misalignment of Optimization Goals:

The paper claims to "unify" two fundamentally distinct generative paradigms: diffusion models, which excel at continuous data generation (typically optimized via ELBO through noise prediction), and autoregressive models, which focus on sequential discrete token prediction (optimized via MLE using cross-entropy loss). This assertion of "unification" at a deep theoretical level is problematic.
- Distinct Mathematical Foundations: Diffusion models learn an iterative denoising process to approximate a data distribution in a continuous space. Autoregressive models, conversely, learn to predict the probability of the next token given all preceding tokens, operating on a discrete vocabulary. These different underlying mathematical formulations and optimization objectives mean they are modeling data generation in fundamentally different ways. Why can more accurate actions be obtained by adding the output together?
- Parameter Sharing vs. Process Unification: While the paper states that both loss functions ($L_{diff}$ and $L_{ce}$ ) "penalize a shared LLM backbone" and their "gradients are jointly backpropagated", this primarily describes a multi-task learning setup with shared parameters. It allows the LLM to learn representations beneficial for both tasks. However, it does not inherently unify the generative processes themselves into a single, cohesive mathematical framework. The continuous nature of diffusion's output ($a_{t+1}^{d}$) and the discrete nature of autoregression's output ($a_{t+1}^{ar}$) before de-tokenization) remain distinct. Any "fusion" occurs at a higher, decision-making level (averaging final actions based on confidence), not at the raw probability or "logits" output layer. The paper's language of "seamlessly incorporating diffusion denoising into the next-token prediction process"  lacks a precise theoretical explanation for how these two fundamentally different generative mechanisms become a single, unified process beyond shared weights.

From my personal perspective, the paper needs to clarify its motivation and further theoretically demonstrate the possibility of hybrid process and why it can work.

---

> ### Author Rebuttal · Authors · 2025-07-31
>
> ## **[W1]. Clarification on Reasoning Capability**
> ### W1-1. LLM's Intrinsic Reasoning Prowess:
> Thank you for your comments. We would like to clarify that we do not claim the reasoning capability of LLMs arises **entirely** from the autoregressive next-token prediction. In Lines 37–43, our intention is to convey that diffusion-based VLA methods [13, 14] leverage the pretrained VLM for multimodal understanding and use it to provide semantically rich latent features for the diffusion head. We did not mean to suggest that diffusion-based VLA completely lack contextual reasoning abilities, but rather that they may miss some degree of reasoning capacity due to (1) the newly injected diffusion head trained from scratch, and (2) the absence of iterative output generation through the next-token prediction paradigm that benefits from the LLM’s internet-scale pretraining.
>
> As you commented, LLM reasoning stems from a combination of factors, including the attention mechanism, large-scale parameters and pretraining, and the autoregressive output paradigm. And our goal is to explore how diffusion-based generation can better utilize the inherent reasoning capabilities of pretrained VLMs. We will improve the writing in the revised version as per suggestion. For example, Lines 40–42 will be updated as follows:
>
>       "Moreover, although diffusion-based VLA models leverage the pretrained VLM to extract semantically rich multimodal latent features as conditions for the diffusion head, the diffusion head itself has not undergone internet-scale pretraining and does not fully exploit the VLM’s original reasoning generation paradigm."
> ### W 1-2. Diffusion Models' Utilization of Context:
> Unlike general large-scale diffusion models [18, 19], current diffusion-based VLA methods struggle to benefit from large-scale pretraining due to the high cost of collecting robotic data. While they can condition on VLM-derived context, they rely on separate diffusion heads trained from scratch. To address this, we introduce a unified VLA paradigm (HybridVLA) that integrates diffusion’s Markovian denoising steps into the LLM’s prediction process via a shared token sequence. This design allows diffusion action generation to inherit the LLM’s pretrained reasoning paradiam, treating each denoising step as an iterative reasoning process.
>
> Furthermore, we conduct additional experiments to validate HybridVLA’s advanced contextual reasoning capabilities. In robotic manipulation, contextual reasoning refers to the policy’s ability to integrate task instructions and visual inputs for closed-loop control. We evaluate this performance on tasks with unseen instructions and novel visual observations.
>
> **For unseen task instructions**, we conduct experiments in the RLBench simulator to evaluate performance under diverse unseen task instructions. Each RLBench task includes multiple semantically equivalent instructions, with different ones used during training and testing. As shown in the table below, while HybridVLA experiences a slight drop in average manipulation success rate, the decrease is smaller than that of other diffusion-based VLA baselines. These results demonstrate HybridVLA’s relatively robust contextual reasoning ability when handling flexible task instructions.
>
> ||Seen task instruction|Unseen task instruction
> -|-|-
> Pi0|0.55|0.45(-18.2%)
> CogACT|0.60|0.52(-13.3%)
> HybridVLA|0.74|0.66(-10.8%)
>
> **For unseen manipulated objects**, as presented in Table 4 of the main paper, we evaluate HybridVLA’s contextual reasoning ability under unseen visual observations. To further validate this, we conduct an additional long-horizon experiment on the “Open drawer & place inside” task. Unlike the visualization in the last row of Figure 5 (Apendix), we modify the test setting by placing not only an orange into the tray but also adding two unseen objects. As shown in the table below, HybridVLA outperforms other methods in handling novel objects, which highlights its capacity for effective reasoning under unfamiliar visual conditions.
>
> ||Orange(Seen)|Charger(Unseen)|Strawberry(Unseen)
> -|-|-|-
> Pi0|0.60|0.50(-16.7%)|0.40(-33.3%)
> CogACT|0.50|0.35(-30%)|0.30(-40%)
> HybridVLA|0.65|0.60(-7.7%)|0.50(-23.1%)
>
> ## **[W2]. Explanation of Optimization Goals and Unification**
> We would like to clarify that HybridVLA does not claim to unify the generation process. Instead, we improve the action generation accuracy by integrating the token sequence formulation and feature representation of two generation methods.
> ### W2-1. Distinct Mathematical Foundations:
> We first illustrate why interegrating these two generation methods can boost feature representation.
>
> **(1) In HybridVLA, the autoregressive and diffusion branches aim to model the same action distribution space** as the action data are normalized in the same way (range [-1,1]). Note that the discrete action is simply a quantized version of the same action distribution. Despite differing in loss formulations, the diffusion and autoregressive branches both aim to approximate the same conditional action distribution, leveraging distinct generative paradigms to model the same data. That is, they both seek to maximize the log-likelihood of the training actions. The autoregressive branch achieves this by explicitly modeling the conditional probability of each action given its preceding context. The diffusion branch models the data generation process as iterative denoising and approximates the log-likelihood of actions by maximizing the ELBO. Despite taking different modeling paths, both branches aim to produce distributional representations that align with the action distributions. Therefore, their imitation learning objectives share the same modeling target, making joint modeling and optimization under shared representations beneficial.
>
> **(2) This joint optimization strategy is also supported by recent unified VLM works, such as Diffusion Forcing [69] (NeurIPS 2024) and Transfusion [70] (ICLR 2025 Oral).** In particular, Transfusion trains a single Transformer architecture that integrates both autoregressive and denoising objectives, enabling generation of discrete text and continuous images. Meanwhile, unlike these prior works, which perform joint modeling over different output modalities, HybridVLA operates in a shared SE(3) action modeling space.
>
> **(3) We further validate, using Principal Component Analysis (PCA) distribution, that jointly optimizing diffusion and autoregressive losses leads to mutual enhancement of their feature representations.**  Specifically, we select several trajectories from the Pick actions and Place actions, and use the corresponding frames as inputs to the model. We then extract both the diffusion-denoised tokens and the autoregressive action tokens, and project them into a 2D space using PCA. We compare the feature distributions produced by models trained with our collaborative training recipe against those from models where each generation branch is trained independently. Training independently means that we apply either the diffusion loss or the autoregressive loss exclusively on our model. As shown in the table below, jointly trained diffusion and autoregressive tokens form tighter intra-class clusters and exhibit larger inter-class distances in both Pick and Place action data. The improved feature representations demonstrate that joint optimization implicitly regularizes the model to retain latent dimensions beneficial for both generation forms.
>
> ||Training collaboratively||Training independently||||
> -|-|-|-|-|-|-
> ||Diffusion token|AR token|\|Independent Diffusion token|\|Independent AR token
> |Intra-class distance|0.49|0.44|\|0.73|\|0.91
> |Inter-class distance|8.7|10.8|\|8.6|\|4.4
>
> ### W2-2. Parameter Sharing vs. Process Unification:
> Regarding “seamlessly incorporating diffusion denoising into the next-token prediction process,” we aim to convey that we integrate the diffusion-denoised tokens into the token sequence of the LLM. This does not imply a unification of the underlying generative paradigms, but a unification of  token sequence formulation of two generation methods with causal attention (L164) .
>
> In HybridVLA, diffusion denoising steps and autoregressive token predictions are embedded within a unified token sequence and jointly processed via causal attention. Instead of treating the two branches as separate heads, the model performs joint modeling within a shared token context. Notably, the denoised action token from diffusion is explicitly used as a condition for the autoregressive prediction, directly influencing the probability distribution of the generated actions. This establishes structural coupling within the LLM’s modeling process, going beyond mere decision-level fusion.
>
>
>
> ## **[Q1]. Misreading of Inference Speed**
> The inference speed values in Table 2 may have been misread: **OpenVLA-7B achieves 6.3 Hz**, while **HybridVLA-7B achieves 6.1 Hz**. The comparison remains fair, as OpenVLA uses a KV-cache for autoregressive generation, and we extend the same mechanism to the diffusion-based iterative denoising process within our token sequence design.
> ## **[Q2]. Additional experiments on SimplerEnv**
> Thank you for your valuable suggestion. We additionally evaluate our model in the SimplerEnv variant aggregation setting using the Google robot. However, since other baselines (e.g., Pi0) are not originally trained on the same subset of the Fractal [86] dataset, we perform consistent fine-tuning across all baselines for fair comparison. Due to time constraints, the remaining results will be included in the revised version.
>
> ||Pick Coke Can|Move Near|Open/Close Drawer|Open Top Drawer and Place|Mean|
> |-|-|-|-|-|-
> |Pi0|0.72|0.50|0.34|0.38|0.49
> |HybridVLA|0.84|0.64|0.40| 0.48|0.59
>
> ---
> **We sincerely thank the reviewer for the detailed comments and valuable time. We hope our responses address your concerns. If any issues remain, please let us know, and we will respond promptly.**

---

> > ### Comment · Reviewer_emK6 · 2025-08-02
> >
> > Thank you for your response. I do not believe it fully addresses my concerns. Here is a point-by-point rebuttal.
> >
> > - In this paper, line 4, "Recent autoregressive vision-language-action (VLA) methods inherit common-sense reasoning capabilities from vision-language models (VLMs) for **next action-token prediction**," and line 31, "autoregressive VLA methods emulate the reasoning paradigm of VLMs for **next token prediction**, effectively leveraging their large-scale pretrained knowledge and reasoning capabilities." The paper clearly states that the authors believe autoregressive VLA models inherit strong common-sense/reasoning abilities from VLMs specifically by using a next-token prediction (NTP) generation method. However, in the rebuttal, the authors attempt to argue that they do not attribute the reasoning capabilities of LLMs to the NTP method, which is a strong contradiction. This misconception is highly relevant to the core motivation of the authors' work, and I do not believe the motivation is sound given this internal contradiction. Furthermore, the authors appear to be deliberately misrepresenting my comments. In the rebuttal, the authors specifically emphasize "not entirely from the autoregressive next-token prediction," while my original review stated, "The paper frequently attributes 'common-sense reasoning' specifically to 'autoregressive generation' and 'next-token prediction.'" I do not feel the authors are answering my question; rather, they are exaggerating my comments.
> >
> > - How should we interpret "the newly injected diffusion head trained from scratch lacks reasoning capability"? For example, do works like RDT/GR00T, which train a diffusion head from scratch, also lack reasoning capabilities? RDT's paper presents a very interesting phenomenon: when the training set only contains instructions to pour 1/3 and pour a full cup of water, the model can still follow an instruction to pour 2/3. Is this an example of reasoning? If so, I believe the authors' argument is invalid. If not, what tasks did the authors use in their paper to demonstrate reasoning?
> >
> > - Is NTP necessary for reasoning? By analogy, in text-to-image tasks, works like [Bagel](https://github.com/bytedance-seed/BAGEL) have also shown that an LLM + Diffusion approach can enable the diffusion head to acquire some reasoning capabilities during image generation. Additionally, the paper mentions that "scene-level semantic reasoning" tasks include "Sweep to dustpan, Water plants, and Frame off hanger." From my understanding of the RLBench paper, I don't see what kind of reasoning capabilities these specific tasks require.
> >
> > - I believe that simultaneously optimizing different representations for the same objective will, to some extent, enhance its representational capabilities, which has been validated in many fields, such as [REPA](https://arxiv.org/abs/2410.06940). However, my main concern is what specific improvements HybridVLA gains during this joint optimization process. How are the discrete tokens learned from the AR objective and the continuous tokens learned from the diffusion objective organically fused? For example, when REPA uses an alignment loss, the model's generation focuses not only on low-level information from the diffusion target but also on high-level semantic information from the alignment target, which is make sense. I do not believe HybridVLA's current design demonstrates this. The authors only claim that continuous action tokens from the diffusion objective are more accurate, while discrete action tokens from the AR objective have more semantic meaning. However, from the perspective of fitting a target distribution, prediction accuracy does not reflect the essential characteristics of the diffusion objective, nor is semantic meaning a feature inherent to autoregressive generation. Based on the authors' current statements, it is difficult to answer my question.
> >
> > - I think this paper is completely unrelated to diffusion forcing or transfusion. The former deals with variable-length diffusion generation, while the latter models different objectives for text/images. The problems they are solving and the technical means they employ are completely different. Transfusion would also **not perform a simple summation of two different distributions modeled under entirely different objectives, such as action embeddings**. The authors should not conflate them.

---

> > > ### Comment · Reviewer_emK6 · 2025-08-02
> > >
> > > - Regarding inference speed, I apologize for mistakenly swapping the frequencies of OpenVLA and Hybrid VLA in my review. This may have caused confusion about the issue itself. I believe that if both models use a KV cache, it is a fair comparison, and I hope this statement can be added to the paper. Furthermore, if $c_{t+1}^{ar} > \theta$ during generation, the HybridVLA model needs to perform a full autoregressive generation of discrete action tokens. Why would this be faster than OpenVLA? I hope the authors can explain this further.
> > >
> > > - For the other supplementary experiment, thank you for the explanation. I have no questions about the experiments on "unseen task instructions" and "unseen manipulated objects."
> > >
> > > Thanks again for the authors rebuttal.

---

> ### Author Response · Authors · 2025-08-03
> **Response to Point 1**
>
> ### Point 1. Revision of the Motivation and Associated Unclear Descriptions
> We sincerely appreciate your prompt response and the time. We also apologize for any remaining concerns and assure you that we had no intention of misrepresenting your comment. However,  due to the rebuttal’s character limit, we are afraid that we didn’t clearly convey our intended idea. Therefore, we’d like to further explain the original intent behind our writing, as well as our motivation for combining diffusion-based and autoregressive generation tokens into a single shared pretrained LLM.
>
> **We completely agree with your perspective** that “reasoning capability is not exclusively dependent on, or solely manifested through, autoregressive next-token prediction.” This is exactly what we intend to convey in both the main paper and the rebutal.
>
> **Revision of related sentences L4 and L31:** However, since this ambiguity has now arisen and your thought means a lot to us, we want to address any potential concerns. To avoid misunderstandings caused by unclear writing, we plan to revise our sentences in Lines 4 and 31 as follows:
>
>     Recent autoregressive vision-language-action (VLA) methods have inherited the pretrained parameters of vision-language models (VLMs) and adopt their next-token prediction output paradigm.
>
> **Reiteration of our motivation:**
> In this paper, our goal is to improve the ability of VLA models to better inherit the pretrained knowledge of VLMs, especially in diffusion-based VLA settings where the VLM serves as the backbone and a DiT module is used as the diffusion generation head, as in Pi0 and CogAct.
>
> We aim to enable diffusion action generation to **more** fully leverage the pretrained knowledge of LLMs, rather than just utilizing the multi modal understanding latent feature from VLMs. Therefore, given the pretrained knowledge of LLMs, we first explore whether directly using large-scale pretrained LLMs as the backbone for diffusion-based action generation is beneficial. Meanwhile, since the iterative output paradigm (NTP) is the original and effective output paradigm of LLMs, we secondly explore whether simulating this output paradigm across the shared token sequence can improve the overall action representation. We will follow your suggestion and improve the related motivation description in the revised version.
>
> **The rationale of our motivation:**
> Through extensive simulation, real-world experiments, and ablation studies in the main paper, we have observed that our approach is indeed effective, providing experimental evidence for its rationale. Additionally, other reviewers have acknowledged the soundness of our motivation; for instance, Reviewer RfcC comments that it “makes sense and is well-motivated.” Meanwhile, Reviewer Ez2J also claims that our framework “is novel and also important for the whole community.”
>
> In conclusion, we would like to reiterate that we are fully aligned with your perspective and strongly agree that "reasoning capability" does not solely originate from NTP. The ambiguity between our views may have arisen due to our unclear writing, which will be carefully addressed in the revised version.

---

> ### Author Response · Authors · 2025-08-03
> **Response to Point 2 & Point 3**
>
> ### Point 2. Diffusion-based VLA possesses reasoning capabilities.
> We fully agree with your perspective. We also acknowledge that existing diffusion-based VLA models possess reasoning capabilities. Based on your constructive suggestion, we will revise the description of diffusion-based VLA models in Lines 40–42 of the main text (e.g., rebuttal W1-1).
>
> In this paper, our objective is to **more** effectively leverage the pretrained knowledge of vision-language models (VLMs) within diffusion-based VLA architectures. Specifically, we aim to: 1) utilize an internet-scale pretrained LLM as the backbone for diffusion-based action generation, and 2) emulate the iterative output generation paradigm inherently adopted by LLMs during large-scale pretraining.
>
> To further compare the design choices of using an LLM as the diffusion backbone versus attaching a separate diffusion head to the end of HybridVLA, we conduct an additional experiment on the RLBench simulator. As shown in the table below, we compare our original HybridVLA with a variant in which a DiT head (with 308M parameters) is appended after HybridVLA. **'HybridVLA-dif' (7B)** integrates diffusion and autoregressive generation during training but relies exclusively on diffusion-based actions during inference. **'HybridVLA+DiT Head'** also utilizes only diffusion-based generation during inference. The results demonstrate that HybridVLA and **HybridVLA-dif** converge faster and enables more robust action generation.
>
>
>
> | | 100 epoch | 150 epoch | 200 epoch | 250 epoch | 300 epoch |
> | --- | --- | --- | --- | --- | --- |
> | HybridVLA+DiT Head | 0.32 | 0.40 | 0.53 | 0.58 | 0.59 |
> | HybridVLA-dif | 0.56 | 0.56 | 0.67 | 0.63 | 0.66 |
> | HybridVLA | 0.63 | 0.68 | 0.72 | 0.69 | 0.74 |
>
> ### Point 3. The Reasoning Capabilities of Diffusion Models and Additional Contextual Reasoning Experiments
> Thank you for your comments. We would like to further express our acknowledgment that diffusion-based VLA models do possess reasoning capabilities. We will revise our description of the relationship between reasoning capability and next-token prediction (NTP) to make it more precise and rigorous in the revised version.
>
> However, regarding Bagel, we would like to clarify that it does not use a separate diffusion head as the backbone for diffusion generation. Instead, it employs either a full LLM (MoT variant) or FFN blocks from the LLM (MoE variant) as the diffusion backbone. Diffusion generation is performed iteratively on the LLM itself. This design aligns with our intuition, as we also aim to use the entire LLM as the backbone for diffusion-based action generation in order to leverage its rich pretrained knowledge and achieve more robust action generation. Meanwhile, we retain the NTP mechanism because LLaMA 2 (our LLM backbone) adopts this generation paradigm during its large-scale pretraining. Thus, we aim to fully leverage the LLM’s original generation process and its inherent capabilities.
>
> In addition to the RLBench simulator, and to better evaluate the contextual reasoning capabilities of our method, we also conduct additional experiments on unseen task instructions and unseen manipulated objects in the rebuttal W 1-2. Our model demonstrates more robust performance than existing diffusion-based VLA methods in handling flexible task instructions and reasoning about unfamiliar visual conditions.

---

> ### Author Response · Authors · 2025-08-03
> **Response to Point 4 & Point 5**
>
> ### Point 4. Effectiveness of our integration design
> **Our design of integrating autoregression and diffusion action tokens in a shared token sequence are shown in Table 1 and Section 3.2**:
> +  We carefully design the integration strategy for diffusion and autoregressive action tokens. We observe that if the autoregressive tokens are placed first, the ground truth (GT) action labels contained in the input during training (a normal learning paradigm in LLM) will leak into the diffusion model’s conditions, negatively impacting the learning of diffusion action tokens. Therefore, we arrange the sequence so that diffusion tokens are placed before autoregressive action tokens. The denoised action token from the diffusion branch is explicitly used as a conditioning signal for the autoregressive prediction, directly influencing the probability distribution of the generated actions.
> + Meanwhile, we introduce special tokens `<BOD>` (beginning-of-diffusion) and `<EOD>` (end-of-diffusion) to encapsulate the diffusion action tokens, clearly distinguishing the boundaries between diffusion and autoregressive action tokens, and ensuring consistent and stable action generation.
> + We further investigate how to encode the robot state and determine its optimal placement within the LLM’s token sequence to enhance temporal consistency in action prediction.
>
>
> **The benefit of our design**:
>
> Empirically, the effectiveness is demonstrated in Table 3. All else being equal, comparing Ex1 and Ex2 (which use only the diffusion branch for test) shows a 6% difference between training with both branches and training with only the diffusion branch. Similarly, comparing Ex3 and Ex4 (which use only the autoregressive branch for test) reveals a 5% difference between training with both branches and training with only the autoregressive branch. The results empirically demonstrate that the Collaborative Training Recipe improves the model’s ability in both diffusion-based action generation and autoregressive generation compared to training them separately.
>
> In analysis, we would like to present our insights into the improvement gains achieved through this collaborative optimization approach:
> + We enable modeling actions at different levels,  namely at the action level and the DoF (degree of freedom) level, to achieve a more robust action representation. **Action-level diffusion action generation:** This manner models all action tokens in parallel by denoising the entire 7-DoF (e.g., single arm) action sequence simultaneously, effectively capturing the overall action dimensions as a whole. This allows for maintaining coherence and integrated action learning across all degrees of freedom. **DoF-level autoregressive action generation:** This manner predicts each DoF token sequentially, paying closer attention to the relationships between individual degrees of freedom within the single action. This allows the model to capture intricate dependencies and constraints between different DoFs. Since both imitation learning objectives share the same target (i.e., action), jointly optimizing them enables the model to learn more robust representations, which in turn mutually enhance the accuracy of both types of action outputs.
>
> + It effectively impacts their feature representation. As shown in the PCA results in W2-1 (3) in rebuttal, collaborative optimization forms tighter intra-class clusters and exhibits larger inter-class distances, both in diffusion and autoregressive tokens. These refined PCA distributions suggest that collaborative training facilitates alignment and mutual enhancement between generation branches, resulting in more structured and robust action representations.
>
> ### Point 5. Why did we mention Transfusion?
> We appreciate your comments. Indeed, unlike **Transfusion**, which focuses on unifying image and text generation, HybridVLA addresses a different problem, which involves collaboratively modeling continuous SE(3) actions for robotic control. However, our motivation for mentioning **Transfusion** in the rebuttal was to illustrate two key similarities with our work: 1) the joint optimization of autoregressive and diffusion losses within a single Transformer model is effective, and 2) diffusion-related tokens can be embedded into the autoregressive token sequence.
>
> While Transfusion does not fuse outputs from different generation objectives, our method integrates both autoregressive and diffusion-based action outputs. This is based on a simple observation in our paper that the two generation outputs exhibit different strengths across manipulation tasks, as shown in Table 7 of the appendix. Based on this observation, we perform adaptive ensemble of the two sets of actions in the SE(3) space, which leads to more robust robot control.

---

> ### Author Response · Authors · 2025-08-03
> **Response to Point 6**
>
> ### Point 6. Inference speed clarification
> We will include this clarification in the revised version of the paper.
> The higher inference speed of **HybridVLA-dif (7B)(9.4hz)** compared to **OpenVLA (7B)(6.3hz)** in Table 2 can be attributed to its inference configuration. As described in Lines 71–73 and 274–276, HybridVLA-dif integrates both diffusion and autoregressive generation during training but relies solely on diffusion-based actions at inference, achieving an inference speed of 9.4 Hz. Specifically, it performs only 4-step DDIM denoising during testing, which results in faster inference than OpenVLA (7B).
>
> In contrast, the **full version of HybridVLA** incorporates the **Collaborative Action Ensemble** mechanism to enhance the robustness of robot control. As you rightly noted, this version performs full autoregressive generation during inference, resulting in a lower inference speed (6.1 Hz) than OpenVLA.
>
>
>
>
>
> **Once again, thank you for all your valuable suggestions and time, which we will carefully address in the revised version. We sincerely hope that our revisions adequately resolve your concerns. If you have any further concerns or questions, please do not hesitate to let us know. We would be happy to provide a prompt and detailed response.**

---

> > ### Author Response · Authors · 2025-08-07
> > **Kind request to continue the discussion with Reviewer emK6**
> >
> > Dear Reviewer emK6,
> >
> > First of all, we truly appreciate your thoughtful feedback and the time you have dedicated to our paper. As the discussion period draws to a close, we sincerely apologize for the interruption and would like to kindly follow up to confirm whether our additional responses have fully addressed your concerns.
> >
> > Specifically, following your valuable suggestions, we have responded to your six latest questions, including: (Point 1) Revised the writing to avoid any potential controversies; (Points 2 & 3) Clarified and added experiments regarding the reasoning capabilities of diffusion-based VLA models; (Point 4) Explained the effectiveness of our integrated and collaborative method design; (Point 5) Clarified the reason for mentioning Transfusion; and (Point 6) Provided further clarification on inference speed.
> >
> > Finally, thank you once again for your detailed comments, which have helped us improve our work. In the revised version, we will incorporate all the proposed experiments and additional explanations, and carefully revise the manuscript to eliminate any potential ambiguities or misunderstandings.
> >
> > Paper 16855 authors

---

> > > ### Comment · Reviewer_emK6 · 2025-08-07
> > >
> > > I find the authors' rebuttal to be unsatisfactory and believe it fails to address the core concerns raised in my initial review. The responses, in fact, have raised further questions and revealed significant inconsistencies in the paper's fundamental motivation and methodology.
> > >
> > > > Point 1:
> > > I do not find the authors' response regarding their core motivation to be a satisfactory explanation. The narrative of the paper's objective has shifted multiple times, and the authors' rebuttal has not resolved this confusion.
> > > - Initial motivation (from the abstract): The reasoning capabilities of autoregressive VLA methods are attributed to Next Action-Token Prediction (NTP), implying that diffusion-based VLA methods, which do not use NTP, inherently lack this ability.
> > > - Subsequent motivation: The claim shifted to "diffusion-based VLA methods lack certain capabilities due to the absence of iterative generation via NTP."
> > > - Current motivation (from the rebuttal): The goal is "to explore shared token-sequence modeling via NTP to improve overall action representation."
> > >
> > > This evolving and inconsistent narrative suggests that the authors themselves may not have a clear understanding of their work's fundamental objective. The final proposed "action ensemble" method appears to be an ad-hoc combination without a clear, consistent rationale. I am not convinced that this approach is driven by the stated goal of inheriting "common-sense reasoning capabilities" from vision-language models.
> > >
> > > Furthermore, I have not seen specific evidence to support the authors' claim that other reviewers found their motivation to be reasonable. On the contrary, other reviews have also pointed out significant conceptual inconsistencies, which I fully agree with:
> > >
> > > - Reviewer RfcC questioned the mathematical justification for tokenizing the entire diffusion chain, citing the Markovian property of the diffusion process.
> > >
> > > - Reviewer cjex asked for a clearer understanding of "contextual reasoning" in the context of manipulation tasks, indicating a lack of clarity on the authors' motivation.
> > >
> > > - Reviewer Ez2J highlighted the fundamental difference between continuous and discrete action spaces, noting that their combination and analysis are not physically or statistically meaningful. I strongly agree with this point.
> > >
> > > > Point 2: Ambiguous "Reasoning Capabilities"
> > >
> > > The authors have failed to address my core question regarding the nature of the "reasoning capabilities" claimed in the paper.  The authors claim that contextual reasoning refers to the policy’s ability to integrate task instructions and visual inputs for closed-loop control. Is that mean change the instruction or change the scene background? I don't think it can represents the "Reasoning Capabilities"
> > >
> > > > Point 3: Misinterpretation of Bagel
> > >
> > > My reference to the Bagel paper was intended to illustrate that LLM+diffusion architectures in other domains have demonstrated reasoning capabilities, thereby questioning the necessity of an NTP-based approach to achieve this. The authors' rebuttal on this point is unclear and does not engage with the intended purpose of my example. I am left with no understanding of their interpretation of my comment or their purpose in discussing Bagel's relevance.
> > >
> > > > Point 4: Unjustified Action Modeling
> > >
> > > I find the authors' explanation regarding different levels of action modeling to be unconvincing.
> > > - The combination of discrete and continuous tokens within the Transformer's causal attention mechanism lacks a clear mathematical or physical justification. These token types do not have an inherent, a priori relationship. To model such a connection meaningfully, one would typically need to introduce a structured kinematic relationship, for instance, by linking joint positions to end-effector pose using a Denavit-Hartenberg (DH) matrix. The current approach lacks this necessary physical awareness.
> > > - The use of PCA visualization on discrete and continuous token features does not provide statistically meaningful evidence to support claims about their semantic relationship.
> > >
> > > > Point 5:
> > >
> > > I am more inclined to believe that this work simply concatenates two disparate training objectives without providing any insight into why this hybrid approach is effective or what novel interactions are being explored. The paper appears to be a technical mash-up rather than a principled exploration of a new model architecture.
> > >
> > > > Point 6: Contradiction Regarding Action Ensemble
> > >
> > > The authors' response on the hybrid-diff version and action ensemble is inconsistent. The method section (lines 245-255) explicitly describes an action ensemble, yet the introduction (lines 71-73) states that the proposed approach does not use ensemble methods. This internal contradiction is a major point of confusion that was not resolved in the rebuttal.

---

> > > > ### Author Response · Authors · 2025-08-08
> > > > **Response to Point 1 from August 7**
> > > >
> > > > ### **Point 1: Further Clarifying the Core Motivation and Addressing Misinterpretations Regarding Reasoning Abilities in Manipulation Tasks**
> > > > **A.** Thank you very much for your further detail comments. Meanwhile, our writing may have unintentionally conveyed viewpoints we did not intend (e.g., as you noted, the **implication** that diffusion-based VLA methods inherently lack this ability without NTP), which led to your main concern. We hope that our further clarification, along with revisions to the relevant parts of the manuscript, will help address your concern.
> > > >
> > > > We would like to clarify that the core motivation of our paper have remained consistent throughout. As explicitly stated in the introduction (Lines 43–45), our core motivation is: "How can we elegantly construct a unified VLA model that seamlessly integrates the strengths of both autoregressive and diffusion policies, rather than simply concatenating them?"
> > > >
> > > > + **Initial statement**: The description of previous autoregressive VLA methods appears in Lines 31-33 of the paper:
> > > > “On the one hand, autoregressive VLA methods [9, 11, 10, 15] emulate the reasoning paradigm of VLMs for next token prediction, effectively leveraging their large-scale pretrained knowledge and reasoning capabilities.”
> > > >
> > > >    We would like to clarify that our statement **should not be interpreted as implying that “diffusion-based VLA methods, which do not use NTP, inherently lack this ability.”** In fact, our intention is simply to describe the respective strengths and limitations of prior autoregressive VLA methods. As clearly stated in our rebuttal, “autoregressive VLA methods inherit the original generation process (NTP) of VLMs,” which we highlight as a strength. However, we also acknowledge their limitation in modeling continuous actions in Line 33-35 of main paper. In contrast to relying solely on autoregressive generation, our actual motivation lies in designing a unified framework that integrates the strengths of both paradigms to improve overall action generation.
> > > >
> > > >
> > > > + **Subsequent discussion and Current motivation**: Again, this is a critique of existing diffusion-based VLA methods. In line 37-43 of paper，we claim that
> > > >   “Recent diffusion-based VLA methods [13, 14, 16, 12] incorporate a diffusion head after the VLM, leveraging probabilistic noise-denoising for action prediction. While these methods enable precise manipulation, **the diffusion head operates independently of the VLM and lacks internet-scale pretraining**. Moreover, **since the head relies solely on VLM-extracted feature representations as input conditions, these methods fail to fully leverage the VLM’s pretrained reasoning capabilities through next-token prediction.**”
> > > >
> > > > Though different in writing, the meaning is consistent with the statement in Lines 5–7 of [Rebuttal by Authors]:
> > > >
> > > > > “ We did not mean to suggest that diffusion-based VLA completely lack contextual reasoning abilities, but rather that they may miss some degree of reasoning capacity due to **(1) the newly injected diffusion head trained from scratch, and (2) the absence of iterative output generation through the next-token prediction paradigm that benefits from the LLM’s internet-scale pretraining.**”
> > > >
> > > >   Moreover, the intended meaning is also aligned with the statement in Lines 14–17 of the [Response to Point 1]：
> > > >
> > > > > “Therefore, given the pretrained knowledge of LLMs, we **first explore whether directly using large-scale pretrained LLMs as the backbone for diffusion-based action generation is beneficial**. Meanwhile, **since the iterative output paradigm (NTP) is the original and effective output paradigm of LLMs, we secondly explore whether simulating this output paradigm across the shared token sequence can improve the overall action representation**.”
> > > >
> > > > Finally, we would like to offer a kindly clarification: **nowhere in the paper do we claim that diffusion-based VLA methods completely lack reasoning capabilities.** Phrases such as "exclusively dependent on," "specifically to," or "solely manifested through" the next-token prediction paradigm, as mentioned in your review, do not reflect our intended meaning. If any part of our discussion on existing VLA methods unintentionally implies such a viewpoint, we sincerely apologize. As stated in our rebuttal, we will revise the wording in the final version to avoid any potential misunderstanding.

---

> > > > ### Author Response · Authors · 2025-08-08
> > > > **Response to Point 4 from August 7**
> > > >
> > > > ### **Point4: Further Clarification of Discrete and Continuous Actions**
> > > > **A.** We are very surprised by your comment that you think discrete and continuous actions “do not have an inherent, a priori relationship”. We would like to clarify that this differs from your example of converting between joint positions and end-effector poses, which requires forward and inverse kinematics. Following OpenVLA, as shown in our rebuttal W2-1 (1), discrete and continuous actions lie in the same action distribution space, since the action data are normalized in the same way (range [−1, 1]). Moreover, a discrete action is simply a quantized representation of that same distribution. In essence, they are just two different expression forms of the same underlying target, either continuous or discrete. Our method simply leverages NTP and diffusion generation to jointly learn actions from this shared distribution. These two generation and modeling paradigms can exist separately, as in autoregressive-based VLA methods or diffusion-based VLA methods, but they can also coexist within a single model, for example, Pi 0.5 KI [3], which uses NTP to generate discrete actions and diffusion to generate continuous actions within the same VLA architecture.
> > > >
> > > > **B.** For the PCA distribution analysis, as shown in our rebuttal W1-2 (3), we apply PCA to the token embeddings of discrete and continuous actions to demonstrate that jointly optimizing diffusion and autoregressive generation leads to a mutual enhancement of their feature representations when facing different actions. This is a practice similar to that in the NeurIPS 2024 paper [4]. Note that, the inputs of our PCA analysis are different actions (i.e., Pick actions and Place actions). Therefore, the results illustrate each branch’s representation of actions, rather than the “semantic relationship”. Moreover, the PCA results show clearer separation and a better understanding of different actions after joint training compared to training each paradigm separately, which strongly suggests that jointly training the two generation paradigms under a unified token sequence with causal attention is both feasible and effective. Similarly, Bagel [1], Transfusion [2], and Pi 0.5_KI [3] demonstrate that joint learning and optimization for NTP-based discrete signals and diffusion-based continuous signals is not only possible but effective.
> > > >
> > > >    We respectfully hope the reviewer may acknowledge that this unified integration strategy aligns with widely adopted practices in recent VLM and VLA research. To avoid potential confusion, we will clarify the direct mapping between discrete and continuous actions more explicitly in the revised version.
> > > >
> > > > [3] VLAs that Train Fast, Run Fast, and Generalize Better
> > > >
> > > > [4] Video Diffusion Models are Training-free Motion Interpreter and Controller

---

> > > > ### Author Response · Authors · 2025-08-08
> > > > **Response to Point 5 from August 7**
> > > >
> > > > ### **Point5: Further Clarification of Our Novel VLA Generation Paradigm**
> > > > First, we would like to clarify that HybridVLA is not a simple “technical mash-up,” but rather a novel exploration of a unified VLA paradigm. As noted by Reviewer RfcC, **“The core idea of joint training of diffusion and autoregressive components makes sense and is well-motivated.”** Reviewer cjex also stated, **“To mitigate interference between the two generation paradigms, this paper proposes a collaborative training recipe that seamlessly incorporates diffusion denoising into the next-token prediction process.”** Similarly, Reviewer Ez2J remarked, **“The whole framework is novel and also important for the whole community,”** and **“I think this paradigm indeed achieves the goal of a generalist model.”** In contrast, for the naive approach of simply concatenating the two generation branches, we provide quantitative results in Table 1 of the main paper, which show a 5%–10% drop in accuracy compared to our carefully designed, robotics-specific unified token sequence.
> > > >
> > > >
> > > >
> > > > In addition to the positive feedback from other reviewers, we would like to reiterate the evolution of the LLM-centric VLA line of work, as it directly highlights the novelty of HybridVLA. The first-generation VLA adopted a straightforward autoregressive approach to predict discretized actions. The second-generation VLA introduced a DiT model conditioned on VLM-extracted features to model continuous actions. However, due to its modular design and focus on generation efficiency, this approach may fail to fully leverage the pretrained knowledge of the LLM for two reasons: (1) the diffusion head is newly introduced and trained from scratch, and (2) it lacks the iterative output process enabled by next-token prediction, which benefits from the LLM’s internet-scale pretraining.
> > > >
> > > > To fully leverage the pretrained knowledge of LLMs within the VLA framework, we began exploring how to condition the diffusion-based action modeling process entirely within the LLM’s original token sequence, thereby enabling a more robust modeling pipeline. This motivated the development of HybridVLA, the first VLA model to integrate diffusion and autoregressive generation within a unified LLM backbone for collaborative action modeling. It frames each denoising step as a reasoning iteration and introduces a robotics-specific unified token sequence to inherit the pretrained reasoning paradigm of the VLM.
> > > >
> > > >
> > > >
> > > > To more clearly highlight our novelty and distinctions from prior VLA work, we provide a detailed summary below:
> > > >
> > > > + **A robotics-specific token sequence formulation is designed to embed diffusion modeling into the LLM’s token stream, leveraging its pretrained knowledge.** This design allows the model to integrate the Markovian denoising steps of diffusion into the LLM’s next-token prediction process. Such integration enables diffusion-based action generation to benefit from the pretrained LLM backbone, treating each denoising step as a reasoning iteration analogous to those in LLMs. Additionally, the diffusion action tokens provide explicit continuous latent conditioning for subsequent autoregressive generation.
> > > > + **A new approach to jointly optimizing unified action representations using diffusion and autoregressive generation.** In HybridVLA, both generative branches aim to approximate the same conditional action distribution, thereby jointly generating distributional representations that align with the action distribution under a shared LLM backbone. This approach differs from general unified VLMs that generate image and text modalities separately, as it facilitates mutually reinforcing optimization and promotes more robust action representation learning.
> > > > + **A collaborative action ensemble mechanism that adaptively fuses both generation branches at test time.** Specifically, we empirically and innovatively observe that combining the two generation branches not only enables mutual enhancement, but also reveals that each branch excels on different tasks or trajectory distributions. As a result, this ensemble mechanism offers a simple yet effective output strategy that leads to more stable control in both simulated and real-world robotic environments.
> > > >
> > > > In summary, we present a novel paradigm for unified VLA action generation, moving beyond simple integration of two generation methods. We hope this summary helps clarify the novelty of our contributions. We will also make these distinctions more explicit in the revised version of the paper.

---

> > > > ### Author Response · Authors · 2025-08-08
> > > > **Response to Point 6 from August 7**
> > > >
> > > > ### **Point6: Further Clarification of HybridVLA and HybridVLA-dif**
> > > >
> > > >
> > > > **We would like to clarify that our statements regarding the use of the action ensemble have been consistent throughout the paper. HybridVLA and HybridVLA-dif are two variants within our proposed framework.** We clearly describe the distinction between these variants whenever they are mentioned, such as in Lines 71–73 and 274–276 of the main paper. Furthermore, the fact that HybridVLA-dif and HybridVLA-ar do not use the collaborative action ensemble is explicitly stated and quantitatively analyzed as part of the ablation study in Lines 302–308 and Table 3.
> > > >
> > > > **Specifically:**
> > > >
> > > > + The method section (Lines 245–255) explicitly describes the action ensemble module, which is a component of the proposed HybridVLA model.
> > > > + In the introduction (Lines 71–73), we state: “To optimize inference speed, we also introduce the HybridVLA-dif (7B) variant, which integrates diffusion and autoregressive generation during training but relies exclusively on diffusion-based actions (without collaberative action ensemble) for inference at 9.4 Hz.” This refers specifically to HybridVLA-dif, an ablation or variant of the full HybridVLA paradigm, which does not employ the action ensemble, as it performs inference solely using diffusion-based actions.
> > > >
> > > > This misunderstanding may be due to our writing, and in the revised version, we will modify Lines 71–73 to:
> > > >
> > > >     “To optimize inference speed, we also introduce the HybridVLA-dif (7B) variant, which integrates diffusion and autoregressive generation during training but relies exclusively on diffusion-based actions (without collaborative action ensemble) for inference at 9.4 Hz.”
> > > >
> > > > ---
> > > > Dear Reviewer emK6,
> > > >
> > > > **Finally, we sincerely thank Reviewer emK6 for the valuable comments and the time dedicated to reviewing our paper. In this work, we indeed propose a novel paradigm for VLA action generation, and we hope our contributions can benefit the broader robotic VLA community. If any part of our writing has unintentionally implied inaccuracies in the discussion of existing work, we will revise it thoroughly in accordance with your detailed suggestions. We genuinely hope you might consider the final positive ratings and comments provided by the other reviewers, and we remain fully committed to addressing all of your concerns to the best of our ability.**
> > > >
> > > > Paper 16855 authors

---

> ### Author Response · Authors · 2025-08-05
> **Further discussion to Reviewer emK6**
>
> Dear Reviewer emK6,
>
> As the discussion period draws to a close, we sincerely apologize for the interruption and would like to kindly check whether our additional responses have addressed your concerns. In line with your suggestions, we will incorporate all the promised experiments and revise the writing to avoid any potential misunderstandings or controversies. Thank you once again for your valuable feedback and time.
>
> Paper 16855 authors

---

> ### Comment · Reviewer_emK6 · 2025-08-07
>
> Based on my review and the authors' rebuttal, my position on this work remains unchanged. The authors have failed to provide convincing explanations for my core concerns, particularly regarding the unclear motivation, lack of methodological rigor, and ambiguous comparisons with existing work. Therefore, I will maintain my score and recommend that the paper be reject.

---

> ### Author Response · Authors · 2025-08-08
> **Response to Point 1 from August 7**
>
> ### **Point 1: Further Clarifying the Core Motivation and Addressing Misinterpretations Regarding Reasoning Abilities in Manipulation Tasks**
>
> **B.** The collaborative "_action ensemble_" strategy is a post-processing strategy designed to produce more accurate action predictions when multiple candidate actions exist at the same timestamp during testing. We introduce this strategy as a natural extension based on the empirical observation that **the two action generation branches (autoregressive and diffusion) exhibit varying strengths across different tasks**. This insight led us to combine their outputs adaptively, and we further validate its effectiveness with an 8% improvement in accuracy, as shown in Table 2 of the main paper. Importantly, the action ensemble operates independently of the model’s common-sense reasoning capabilities. Its purpose is solely to refine the final action outputs at inference time, not to enhance or contribute to reasoning. We would like to clarify that **at no point in the paper or rebuttal do we claim that this ensemble method has anything to do with common-sense reasoning**.
>
> **C.** **Though other reviewers stated these problem during the first round rebutal, they all claim now that we have solved their concerns, meaning these problems no longer exists after our rebuttal**. Specifically：
>
> >The concern raised by Reviewer RfcC is primarily due to our Figures 1 and 2, as well as insufficient detail in our description of diffusion token modeling. The reviewer initially misunderstands our method as “feeding the entire diffusion chain as tokens into the LLM.” However, once we clarify that only a single noisy sample from the current denoising step is fed into the LLM to predict the corresponding noise, **the reviewer responds that this explanation “addressed my major concerns.”** Therefore, we believe this is a misunderstanding caused by our unclear presentation, not an issue of mathematical soundness or justification. At the same time, Reviewer RfcC’s comment will help us refine the description of diffusion token modeling.
>
> >The "contextual reasoning" concern raised by Reviewer cjex has been addressed through our clearer definitions and the addition of new experiments involving unseen instructions and unseen objects from different semantic categories (long-horizon task). **Notably, Reviewer cjex responds that “most of my concerns are well addressed.”** We sincerely appreciate Reviewer cjex for raising this important point, as it motivates us to further validate the effectiveness of our method and demonstrate its performance in contextual reasoning within manipulation tasks.
>
> >The concern raised by Reviewer Ez2J has also been addressed by providing additional experiments demonstrating the effectiveness of both autoregressive and diffusion-based action generation, along with the corresponding PCA analysis. **Reviewer Ez2J responds, “I think this paradigm indeed achieves the goal of a generalist model.”** We are also sincerely grateful to Reviewer Ez2J for raising this point, which prompted us to further validate the effectiveness of our motivation and method through PCA distribution analysis.
>
>
>
> **Finally, we would be deeply grateful if you could consider revisiting your evaluation based on our responses, as well as the rebuttal feedback and positive final ratings from the other reviewers. Meanwhile, our writing may have unintentionally implied viewpoints that we did not intend to express (e.g., as you noted, “implying that diffusion-based VLA methods, which do not use NTP, inherently lack this ability”), which led to your main concern. Please rest assured that we will revise the relevant parts of the manuscript in the revised version to address the issues you pointed out.**

---

> ### Author Response · Authors · 2025-08-08
> **Response to Point 2 and Point3 from August 7**
>
> ### **Point 2: Reasoning Capabilities in Manipulation Tasks**
>
> First, regarding the definition of "reasoning" ability in manipulation tasks, we, along with Reviewer cjex and Reviewer Ez2J, agree that a VLA model’s capacity to **integrate flexible unseen task instructions and generalize to unseen visual configurations in long-horizon tasks while still generating relatively accurate actions** can reasonably be interpreted as a form of reasoning in robotic manipulation.
>
> Moreover, in your constructive response on August 2 (lines 10–13), you commented that:
> “How should we interpret 'the newly injected diffusion head trained from scratch lacks reasoning capability'? For example, do works like RDT/GR00T, which train a diffusion head from scratch, also lack reasoning capabilities? RDT's paper presents a very interesting phenomenon: when the training set only contains instructions to pour 1/3 and pour a full cup of water, the model can still follow an instruction to pour 2/3. Is this an example of reasoning?”
>
>
> We completely agree with your point that RDT demonstrates the contextual reasoning ability of DiT-based policy models, using examples such as changing the task instruction from “pour 1/3 cup of water” (seen during training) to “pour 2/3 cup” (unseen). In a similar spirit, we believe our experiments, where we modify task instructions from seen to unseen variants and manipulate objects from seen to unseen ones (belonging to different semantic categories), **can also evaluate the model’s contextual reasoning capability.** For example, during training, our HybridVLA has **never encountered tasks like “pick up the charger/strawberry and put it in the drawer.”** Besides, as shown in our rebuttal (W1–2), we conducted an additional long-horizon experiment on the “Open drawer & place inside” task, which **not only introduces unseen objects during training but also increases the overall number of steps and atomic subtasks.**
>
> ---
>
> ### **Point3: Misinterpretation of Bagel**
> Thank you for your further comments. We apologize if our response gave the impression that we were discussing Bagel's relevance.
> **Our response aims to clarify that Bagel employs both NTP and diffusion modeling within an unified model simultaneously, which is entirely consistent with our approach. Therefore, we would like to clarify that Bagel serves as supporting evidence for the necessity of an NTP-based method and our concept of collaborative NTP and diffusion generation**
>
> As you noted in reference to the Bagel [1] model, “works like Bagel have also shown that an LLM + Diffusion approach can enable the diffusion head to acquire some reasoning capabilities during image generation.” As demonstrated by Bagel and other studies [2], jointly modeling autoregressive and diffusion-based outputs within a unified backbone can both fully leverage the pretrained knowledge of the LLM and enable the diffusion branch to acquire nontrivial reasoning abilities.
>
> This is precisely the foundation of our work. Our goal is to maintain the autoregressive pathway's capacity for action generation while integrating the diffusion-based module to enhance continuous action modeling. Critically, we explore a collaborative training paradigm and token-sequence design that ensures the two branches do not interfere with each other, but instead reinforce one another. More importantly, as shown in the ablation study (Table 3, Ex1 and Ex2), we empirically validate that collaborative NTP-based action generation and diffusion-based action generation during training can enhance the accuracy of the diffusion branch’s action generation during testing.
>
> Finally, we respectfully offer a gentle clarification: Bagel is not a model that combines an LLM with an external diffusion head; instead, it adopts a unified architecture that integrates both autoregressive and diffusion-based generation.
>
>
> [1] Emerging Properties in Unified Multimodal Pretraining
>
> [2] Transfusion: Predict the Next Token and Diffuse Images with One Multi-Modal Model

---

### Official Review · Reviewer_Ez2J · 2025-07-02

**Clarity:** 3
**Significance:** 4
**Originality:** 4
**Rating:** 5
**Confidence:** 5

**Summary:**

This paper proposes a novel VLA framework, HybridVLA, a unified approach that combines the continuous nature of diffusion-based actions with the contextual reasoning capabilities of autoregression. To ensure its effectiveness, it employs a collaborative training recipe (CTR), which integrates diffusion denoising into the next-token prediction process. Extensive experiments on both simulated and real-world scenarios demonstrate a higher success rate compared to state-of-the-art (SOTA) methods.

**Questions:**

Please refer to the weaknesses.

**Ethical Concerns:**

["NO or VERY MINOR ethics concerns only"]

**Final Justification:**

This is the first work that combines AR and Diff  approaches in VLA, ensuring that while AR provides strong semantic understanding, the Diff paradigm can still be effectively leveraged for fine-grained action refinement. This leads to a new VLA architecture that benefits from both methods. From the perspective of the entire community, this architecture pushes the development of model structures from solely AR or AR+policy head setups towards a more unified architecture. Therefore, it holds significant importance.

During the rebuttal process, the authors addressed my initial confusion about the intuition behind designing such an architecture. I am inclined to accept this paper.

**Quality:**

4

**Strengths And Weaknesses:**

Strengths:
1. The whole framework is novel and also important for the whole community.
2. The experiments on simulation and real-world scenarios are extensive, solid and convincing.
3. The ablation studies on the confidence threshold, the impact of the KV cache, and the impact of denoising steps greatly enhance the richness of the article.

Weakness:
1. Regarding lines 58–59, the author emphasizes that the tasks applicable to AR and Diffusion are different, and this is also demonstrated in supplementary experiments. However, this explanation doesn't sound very intuitive. From my understanding, is it the case that the features used for diffusion generation are different from those used for AR generation? Could the author explain this and provide additional experiments to clarify?

2. For diffusion, the generated actions are already continuous, so why is an autoregressive model needed to generate discrete actions? Perhaps the action space itself is different. If that's the case, would my understanding be correct that switching from the epsilon prediction paradigm to the v prediction paradigm could potentially solve this issue?

3. Diffusion models can generate with continuous chunking, but autoregressive models find it challenging to achieve chunking for actions, which may affect potential efficiency.

---

> ### Author Rebuttal · Authors · 2025-07-31
>
> ## **[W1]. The difference between AR and Diffusion action generation**
>
> Thank you for your thoughtful comments. The statement in Lines 58–59 is based on empirical findings from the experiment in Appendix B.1 (Table 7), where diffusion-generated actions and autoregressive-generated actions exhibit varying performance across tasks. To provide a more intuitive explanation of this observation, we elaborate from three perspectives: (1) differences in generation mechanisms and conditioning features, (2) a Principal Component Analysis (PCA)-based distribution analysis, and (3) additional experiments on unseen manipulated instances.
>
> ### A. Differences in generation mechanisms and conditioning features:
> Unlike the autoregressive action generation approach, which requires quantizing actions into discrete bins, the diffusion branch can directly generate continuous actions. HybridVLA performs multi-step Markovian denoising within the LLM, treating each step as a reasoning iteration. This process enables HybridVLA to progressively refine action predictions by leveraging the pretrained knowledge of the LLM. **As a result, diffusion-based generation is particularly suitable for producing smooth and fine-grained action sequences, which are essential for high-precision manipulation tasks.** In contrast, the autoregressive branch can condition not only on task instructions and visual observations but also on the denoised tokens produced by the diffusion branch. These multimodal conditions and diffusion action latent conditions enable the autoregressive branch to reason over a broader context, making it more effective at capturing task-dependent semantic alignment between the environment and action intent.
>
>
>
> ### B. Principal Component Analysis (PCA):
> To more robustly evaluate the representational differences between the two generation branches across different tasks, we apply Principal Component Analysis (PCA) for dimensionality reduction and compute the intra-class and inter-class distances in the principal component space. Specifically, we select several trajectories from the Pick actions (real-world single-arm setup) and modify only the manipulated object (e.g., replacing a cube with a charger or a strawberry), while simultaneously updating the corresponding language instruction. We then extract the diffusion-denoised tokens and the autoregressive action tokens, and project them into 2D using PCA to obtain their distributions across three different object variants. We observe that the diffusion-based features show relatively blurry clustering in PCA space across different object categories. In contrast, the autoregressive action token embeddings form tighter intra-class clusters and exhibit larger inter-class margins, indicating greater robustness to the semantic differences in manipulated objects. This result supports our empirical observation that autoregressive action representations capture relatively richer semantic context. Due to rebuttal constraints that prohibit including additional visualizations, we instead report the intra-class and inter-class distances of the two generation paradigms in the table below.
>
> |  | Diffusion token | AR token |
> | --- | --- | --- |
> | Intra-class distance | 1.24| 0.78|
> | Inter-class distances |7.37 | 11.60|
>
>
>
>
> ### C. Additional unseen instances manipulation experiments:
> For additional experiments on unseen manipulated instances, we follow the Pick and Place task setting from the generalization experiment (Section 4.4) using a single-arm robot. Specifically, we modify the manipulated object to previously unseen objects, such as a charger and a strawberry. As shown in the table below, HybridVLA-AR exhibits a smaller performance drop after object replacement. This further supports our claim that the autoregressive branch is more effective at capturing semantic variations.
>
>
>
> |  | Seen object | Unseen objects | Drop percentage |
> | --- | --- | --- | --- |
> | HybridVLA-ar | 0.80 | 0.55 | 31.3% |
> | HybridVLA-dif | 0.85 | 0.50 | 41.2% |
>
> ---
> ## **[W2]. Why is autoregressive discrete action generation essential in HybridVLA**
>
> **First, diffusion-based and autoregressive action generation exhibit different strengths across tasks,** as evidenced by the empirical findings in Appendix B.1 (Table 7). This suggests that the differences are not solely attributable to their output formats (continuous vs. discrete), but also to their fundamentally distinct generation paradigms. Specifically, as discussed in W1.B and W1.C, our PCA-based distribution analysis and additional manipulation experiments both indicate that the autoregressive branch is more effective at capturing semantic variations when handling unseen visual configurations.
>
> **Second, the combination of diffusion and autoregressive generation yields a more stable optimization objective.** The two generation branches both aim to approximate the same conditional action distribution, leveraging different generative paradigms to model the same observed data. That is, they both seek to maximize the log-likelihood of the training actions. Therefore, introducing autoregressive generation during training promotes more robust action representation learning. Our ablation results (e.g., Ex1 vs. Ex2, Ex3 vs. Ex4) further validate this, showing that jointly trained diffusion and autoregressive branches outperform their independently trained counterparts. This validates that retaining autoregressive discrete action generation within the hybrid objective serves as a mutual regularizer and enhances action modeling. As per your valuable suggestion, in the future version, we will switch from the epsilon prediction paradigm to the v prediction paradigm to explore whether it can lead to more robust collaborative action generation.
>
> ---
> ## **[W3]. Action chunking**
>
> Good question. There are two ways to incorporate action chunking. (1) We can apply action chunking scheme to the diffusion-based action generation, while the autoregressive branch only predicts the next immediate action. In this setup, the Collaborative Action Ensemble strategy is applied only at the most critical first step, while the subsequent actions are solely adopted from the diffusion-generated action chunk. This approach preserves the entire efficiency of action generation. (2) Additionally, we can adopt parallel decoding to accelerate the generation of action chunks in the autoregressive branch as much as possible. This enables the Collaborative Action Ensemble strategy to operate over the entire action chunk.
>
> ---
> **Finally, we sincerely thank the reviewer for the thoughtful comments and valuable time. We hope our detailed responses address your concerns. If any issues remain, please let us know, and we will respond promptly.**

---

> > ### Comment · Reviewer_Ez2J · 2025-08-02
> > **Official Comment by Reviewer Ez2J**
> >
> > Thank you for the authors’ response, which addressed most of my concerns, especially regarding the necessity of combining AR and DP to ensure semantic generalization while retaining accurate action execution.
> >
> > However, I still have a minor question. Could the authors provide stronger evidence or a specific example demonstrating that a Diffusion-based decoding strategy can outperform AR in fine-grained or dynamic manipulation tasks? If such evidence can be presented, it would further support the aforementioned viewpoint.
> >
> > If the author could address this issue, I am willing to raise my score for the paper.

---

> > > ### Author Response · Authors · 2025-08-03
> > > **Response to Reviewer Ez2J’s Further Comment and Additional Experiments**
> > >
> > > We are pleased to hear that our response has addressed most of your concerns. We also sincerely appreciate your constructive follow-up comments. Following your suggestion, we conduct additional fine-grained and dynamic manipulation experiments to evaluate the advantages of diffusion-based actions using the AgileX dual-arm robot.
> > >
> > > ## 1. Additional Evaluation
> > > ###  a. For the fine-grained manipulation task.
> > > Due to time constraints, we collect 30 new trajectories involving dual-arm collaboration to unplug a charging cable from its docking base. These demonstrations are acquired via master-puppet teleoperation, and all other experimental settings remain consistent with those described in the dual-arm experiment section (main paper). In this task, we evaluate both **HybridVLA-dif** and **HybridVLA-ar**, with the results presented in the table below. Both models integrate diffusion and autoregressive generation during training; however, **HybridVLA-dif** relies exclusively on diffusion-based actions, while **HybridVLA-ar** relies exclusively on autoregressive actions during inference.
> > >
> > >
> > > |  | Successful rate |
> > > | --- | --- |
> > > | **HybridVLA-dif** | 0.60 |
> > > | **HybridVLA-ar** | 0.20 |
> > >
> > >
> > > As described in our rebuttal, the diffusion branch directly generates continuous actions and embeds multi-step Markovian denoising within the LLM, treating each step as a reasoning iteration. This iterative process fully leverages the LLM’s pretrained knowledge to progressively improve action predictions. Consequently, diffusion-based generation exhibits greater robustness in fine-grained manipulation tasks.
> > >
> > > ### b. For the dynamic manipulation task.
> > > We conduct additional evaluations using the pick-and-place setup. We directly load the trained model and test it by dynamically moving the banana on the table before the pick-up action, as shown in the first row of Figure 6. Note that the banana’s movement range remains within the maximum manipulable scope of the left arm, and its position is periodically shifted left and right. In this task, we also evaluate both **HybridVLA-dif** (using only diffusion-based action generation) and **HybridVLA-ar** (using only autoregressive action generation), with the results presented in the table below.
> > >
> > > |  | Successful rate |
> > > | --- | --- |
> > > | **HybridVLA-dif** | 0.75 |
> > > | **HybridVLA-ar** | 0.55 |
> > >
> > >
> > >
> > >
> > > The results indicate that **HybridVLA-dif** also exhibits an accuracy gain over **HybridVLA-ar** in dynamic object manipulation scenarios.

---

> > > > ### Comment · Reviewer_Ez2J · 2025-08-04
> > > > **Comment by Reviewer Ez2J**
> > > >
> > > > Thank you for your response regarding my concerns about the practical performance in fine-grained and dynamic experiments.
> > > >
> > > > I think this paradigm indeed achieves the goal of a generalist model: ensuring semantic generalization while retaining accurate action execution through the proposed hybrid model architecture and training approach.
> > > >
> > > > I am inclined to recommend acceptance.

---

> > > > > ### Author Response · Authors · 2025-08-04
> > > > >
> > > > > Dear Reviewer Ez2J,
> > > > >
> > > > > Thank you very much for recognizing our work and for the positive rating. We will include the additional explanations and experiments in the revised version.
> > > > >
> > > > > Paper 16855 authors

---

### Official Review · Reviewer_cjex · 2025-07-02

**Clarity:** 3
**Significance:** 3
**Originality:** 3
**Rating:** 4
**Confidence:** 4

**Summary:**

To leverage the VLM’s pretrained reasoning capabilities, this paper introduces HybridVLA, a unified framework that absorbs the continuous nature of diffusion-based actions and the contextual reasoning of autoregression within a single large language model. HybridVLA outperforms previous state-of-the-art VLA methods by 14% and 19% in mean success rate on simulation and real-world tasks, respectively, while demonstrating stable manipulation in unseen configurations.

**Questions:**

See weakness

**Ethical Concerns:**

["NO or VERY MINOR ethics concerns only"]

**Final Justification:**

Thank the authors for their responses to my concerns. Most of my concerns are well addressed. Hence, I will maintain my score as borderline accept.

**Limitations:**

One limitation of HybridVLA is that its inference speed is constrained by the slower autoregressive generation, similar to prior autoregressive VLA methods

**Quality:**

3

**Strengths And Weaknesses:**

To mitigate interference between the two generation paradigms, this paper proposes a collaborative training recipe that seamlessly incorporates diffusion denoising into the next-token prediction process. However, here are my concerns:

1.	How to understand the contextual reasoning of autoregression in manipulation tasks? Additionally, in experiments, how to demonstrate that the proposed model outperforms comparative methods in this aspect?

2.	This paper omits discussions on several recent studies regarding diffusion-based VLA models, such as [1] and [2].

3.	In the framework of HybridVLA, the selection of parameter $\theta$ exerts a substantial influence on the model's output. How did the authors determine the specific value of $\theta$?

4.	To validate the performance of the proposed model and enhance reproducibility, it is recommended that the authors conduct generalization experiments in a simulation environment, such as [3].

[1] Fine-tuning vision-language-action models: Optimizing speed and success

[2] π_{0.5}: a Vision-Language-Action Model with Open-World Generalization

[3] SimplerEnv: Simulated Manipulation Policy Evaluation Environments for Real Robot Setups

---

> ### Author Rebuttal · Authors · 2025-07-31
>
> ## **[W1]. Contextual Reasoning Comparison with Baseline Methods in Manipulation Tasks**
>
> ### A. Definition of context reasoning for VLA:
>
> Thank you for your constructive comments. In robotic manipulation tasks, contextual reasoning for VLA models refers to the policy’s ability to integrate task instructions and robotic observations in order to generate closed-loop action sequences. HybridVLA improves this by leveraging a unified token sequence and a corresponding collaborative training recipe that integrates the Markovian denoising steps of diffusion into the next-token prediction process, where each denoising step is treated as a reasoning iteration. This design not only enables HybridVLA to support both diffusion-based and autoregressive action generation, but also inherits the reasoning paradigm of large-scale pretrained VLMs.
>
> ### B. Additional context reasoning experiments:
>
> Following your valuable suggestion, to empirically evaluate contextual reasoning, we conduct two additional experiments, including inference on unseen task instructions and unseen visual observations.
>
> (1) **For unseen task instructions**, we conduct additional experiments in the RLBench simulator to evaluate performance under diverse unseen task instructions. Each task in RLBench is associated with multiple semantically similar instructions, but during training, both HybridVLA and all baselines are only exposed to the longest version. At test time, we randomly select one of the previously unseen instructions for each rollout. As shown in the table below, although HybridVLA shows a slight performance drop compared to its results in Table 2 of the main paper, the decrease is significantly smaller than that of other VLA baselines. This demonstrates HybridVLA’s more robust contextual reasoning ability for diverse task language instructions.
>
>
>
> |  | Seen task **instruction** | Unseen task **instruction** |
> | --- | --- | --- |
> | Pi0 | 0.55 | 0.45 (-18.2%) |
> | CogACT | 0.60 | 0.52 (-13.3%) |
> | HybridVLA | 0.74 | 0.66 (-10.8%) |
>
> (2) **For unseen manipulated objects**, as shown in Section 4.4 and Table 4 of the main paper, we assess HybridVLA’s contextual reasoning ability when faced with unseen observation configurations. To further validate this, we conduct an additional long-horizon experiment on the “Open drawer & place inside” task, in which three objects are placed on the plate, two of which belong to unseen categories during training. Specifically, unlike the visualization in the last row of Figure 5 in the Appendix, we modify the test setting by placing not only an orange into the tray but also adding two unseen objects, including a charger and a strawberry. Since visualizations are not allowed in the rebuttal, a similar experimental procedure can be referenced around the 5:10 timestamp in the submitted supplementary video. As shown in the table below, HybridVLA outperforms other methods in handling novel objects, especially in the long-horizon task, highlighting its ability to reason effectively under unfamiliar visual conditions.
>
> |  | Orange (Seen) | Charger (Unseen) | Strawberry (Unseen) |
> | --- | --- | --- | --- |
> | Pi0 | 0.60 | 0.50 (-16.7%) | 0.40 (-33.3%) |
> | CogACT | 0.50 | 0.35 (-30%) | 0.30 (-40%) |
> | HybridVLA | 0.65 | 0.60 (-7.7%) | 0.50 (-23.1%) |
>
> ---
> ## **[W2]. Discussion and Comparison with Recent VLA Models [1,2]**
>
> Thank you for your suggestion. We will add the discussion and comparison in the revised paper as follows.
>
> ### A. Discussion with [1, 2]:
> Pi0.5 [2] aims to build diffusion-based VLA models with a separate diffusion head, while OpenVLA-OFT [1] similarly adopts an independent MLP head for either regression-based or diffusion-based action generation. However, our HybridVLA adopts a unified LLM to perform collaborative diffusion and autoregressive action generation, which differs from these approaches in both implementation strategies and optimization objectives.
>
> In detail, **Pi0.5** extends the modular architecture of Pi0 [13] by adding an action expert module following the VLM, and introduces additional training data with intermediate language instructions. This enables the model to first generate low-level commands and subsequently predict actions conditioned on them, offering greater flexibility in decoupling high-level semantics from low-level execution.
>
> **OpenVLA-OFT** proposes a parallel decoding strategy with action chunking, which generates continuous actions via the MLP head. This design substantially enhances inference efficiency, enabling the model to operate at high frequencies.
>
> Unlike [1, 2], **HybridVLA** proposes using a shared LLM as the backbone to construct a unified token sequence that integrates both diffusion and autoregressive action generation paradigms. We integrate the Markovian denoising steps of diffusion into the LLM’s next-token prediction process, treating each step as a reasoning iteration. This results in a new action modeling paradigm in which the two generation mechanisms can mutually enhance each other. The following is a summary table comparing the three methods.
>
>
> | | Diffusion action generation  | AR action generation | Diffusion module |
> | --- | --- | --- | --- |
> | Hybrid-VLA | yes | yes | LLM |
> | Pi-0.5 | yes | yes | Additional DiT Head |
> | Openvla-OFT | yes | no | Additional MLP Head |
>
>
> ### B. Comparison with [1]:
> As shown in the table below, since Pi0.5 has not been open-sourced, we conduct quantitative comparisons only with OpenVLA-OFT on the RLBench simulator. Using the same dataset, we follow the official simulation fine-tuning setting provided by OpenVLA-OFT for training. Our method outperforms OpenVLA-OFT in manipulation accuracy by fully leveraging the two generation mechanisms and the pretrained knowledge of the LLM.
>
> | Method                     | Mean S.R. |
> |----|---|
> | OpenVLA-OFT (regression)  |    0.45       |
> | HybridVLA                 | 0.74      |
>
> ---
> ## **[W3]. The selection of parameter θ**
> To determine the value of the parameter θ (confidence threshold), we have conducted large-scale empirical experiments. As shown in Lines 249–251, we find that for our autoregressive action generation, over 80% of successful trajectories had an average action token confidence exceeding 0.96. Meanwhile, as shown in Appendix B.2 and Table 8 (Lines 792–803), we conduct an ablation study on different threshold values. The results confirm that setting the threshold to 0.96 yields the most effective fusion between the two action branches, leading to more robust robot control.
>
> Moreover, the choice of this hyperparameter remains effective in real-world robot experiments. Since HybridVLA is pre-trained on extensive real-world robotic data, we observe that when the confidence of an autoregressive action token exceeds 0.96, the corresponding action is highly likely to succeed during real-robot deployment.
>
> ---
> ## **[W4]. Additional Generalization Experiments in Simulation**
>
> Thank you for your valuable suggestion. We additionally evaluate our model in the SimplerEnv variant aggregation setting using the Google robot. As shown in the table below, our method achieves satisfactory generalization performance. However, since other baselines (e.g., Pi0) are not originally trained on the same subset of the Fractal [86] dataset, we conduct consistent fine-tuning across all baselines using the same dataset to ensure a fair comparison. Due to time limitations, we complet only the comparative experiments for Pi0. The remaining results will be included in the revised version.
>
> | Method     | Pick Coke Can | Move Near | Open/Close Drawer | Open Top Drawer and Place | Mean S.R. |
> |----|---|---|----|---|---|
> |Pi0|0.72|0.50|0.34|0.38|0.49
> |HybridVLA|0.84|0.64|0.40| 0.48|0.59
>
> ---
> **Finally, we sincerely thank the reviewer for the thoughtful comments and valuable time. We hope our detailed responses address your concerns. If any issues remain, please let us know, and we will respond promptly.**

---

> > ### Comment · Reviewer_cjex · 2025-08-07
> > **Thank the authors for their responses**
> >
> > Thank the authors for their responses to my concerns. Most of my concerns are well addressed. Hence, I will maintain my score as borderline accept.

---

> > > ### Author Response · Authors · 2025-08-07
> > >
> > > Dear Reviewer cjex
> > >
> > > **We are truly delighted to have addressed most of your concerns, and we sincerely appreciate your positive rating.** In the revised version, we will incorporate all additional contextual reasoning experiments, comparisons with recent VLA models, and simulation generalization experiments into the main paper. We will also include additional explanations regarding the rationale behind the selection of parameter θ. Finally, thank you once again for your valuable comments, which have further enhanced the completeness of our work.
> > >
> > > Paper 16855 authors

---

> ### Author Response · Authors · 2025-08-04
> **Further discussion to Reviewer cjex**
>
> Dear Reviewer cjex,
>
> Thank you for the time and effort you dedicated to reviewing our paper. We have provided point-by-point responses to the raised questions, including the definition and additional experiments on contextual reasoning in manipulation tasks (W1), discussions and comparisons with several recent studies [1,2] (W2), an explanation of the parameter selection rationale (W3), and generalization experiments conducted in SimplerEnv (W4).
>
> As the rebuttal period has now passed the halfway point, we hope our responses have addressed your concerns and would be grateful if you could consider updating the score should you find our clarifications satisfactory. We remain available for any further discussion if there are unresolved concerns. Thank you again for your thoughtful feedback and positive recognition of our work.
>
> Paper 16855 authors

---

### Official Review · Reviewer_RfcC · 2025-07-03

**Clarity:** 3
**Significance:** 2
**Originality:** 3
**Rating:** 4
**Confidence:** 2

**Summary:**

The authors propose a method for jointly training the diffusion and autoregressive components of a Vision-Language-Action (VLA) model. Rather than training the autoregressive component first and the diffusion component separately—as is common in prior work—the authors present a unified architecture and training process. This approach enables more effective collaboration between the two components and leads to improved success rates across multiple robotic manipulation tasks.
Strengths and Weaknesses

**Questions:**

Why is it necessary to feed in the entire diffusion chain? Isn’t the final action a sufficient statistic for the distribution, especially given that earlier samples just introduce additional variance?

Is appending the diffusion sequence primarily a convenience or is there a theoretical or empirical justification for this design choice?

**Ethical Concerns:**

["NO or VERY MINOR ethics concerns only"]

**Final Justification:**

The authors addressed all of my previously stated concerns, and I have no further major issues about the paper.

**Limitations:**

yes

**Quality:**

3

**Strengths And Weaknesses:**

Quality:

The paper presents a method that achieves better success rates on several tasks and the highest average success rate overall.

The core idea of joint training of diffusion and autoregressive components makes sense and is well-motivated.

Some important design choices (e.g., appending the entire diffusion chain) lack justification, and it's difficult to understand exactly how the integration is implemented.

Clarity:

The writing is generally clear, but the technical details of the method are hard to follow in places.

It's not really clear why the entire diffusion chain is included as tokens, instead of predicting one at a time. Both the forward and backward diffusion processes are Markovian, so the current noised action should always be a suffient statistic for future action predictions. It's unclear if there is a mathematical justification for why the diffusion chain prediction is set up the way it is.

Significance:

It’s useful to have a demonstration that joint training of these components is possible and improves performance.

However, the significance is somewhat limited because it builds on ideas already explored in related works (e.g., $\pi_0$, Diffusion Forcing [1]).

One nice takeaway takeaway appears to be that wrapping the diffusion sequence in BOD and EOD tokens is enough to ensure consistency and stability

Originality:

The idea of combining LLM-style token prediction with diffusion models is common, and other works have also considered joint training of diffusion and autoregressive components. The novelty here lies in adapting this idea specifically to the VLA setting, using a pre-trained LLM.

---

> ### Author Rebuttal · Authors · 2025-07-31
>
> ## **[W1,Q1,Q2] Clarification on diffusion chain modeling in HybridVLA**
>
> ### A. Clarification of the implementation:
>
> Thank you for the insightful question. **We would like to clarify that HybridVLA does not feed the entire diffusion chain as tokens into the LLM. Only one noisy sample from the current denoising step is fed into the LLM to predict the corresponding noise.** Below, we provide a detailed explanation of our implementation.
>
> (1) During training, as shown in Lines 177–180, we follow the standard DDPM training scheme by randomly sampling a single denoising timestep and the corresponding noisy action, which are projected into the LLM’s token embedding space.
>
> (2) During inference, as shown in Lines 232–234, HybridVLA adopts a step-wise DDIM denoising process, where at each step, only the current noisy sample is fed into the LLM to predict the noise. The predicted noise is then subtracted from the noisy
> sample to produce the input for the next step, while the token sequence does not retain any previous noise samples. This design enables multi-step Markovian denoising within the LLM, where each step is treated as a reasoning iteration. In this way, HybridVLA can progressively refine action predictions by leveraging the LLM’s large-scale pretrained knowledge.
>
> Importantly, as you correctly pointed out (“earlier samples just introduce additional variance”), our method does not feed the entire diffusion chain into the LLM, while only processing the current noisy token at each step.
>
> ### B. Clarification of Figures 1 and 2:
>
> Figures 1 and 2 aim to illustrate that each denoised sample is reused as input for the LLM’s prediction, forming an iterative process. However, the static illustrations may misleadingly suggest that the entire diffusion sequence is input at once. **Based on your valuable comments, we will revise both the figures and the corresponding descriptions to clarify the denoising mechanism.**
>
> ### C. Empirical justification:
>
> To further validate our diffusion design choice, we conduct an additional experiment in which we explicitly included earlier noise samples in the token sequence. Note that this process retains only the previously input noisy sample, but does not perform parallel denoising over the entire noise chain. We follow the same collaborative training recipe as in the main paper and report results for HybridVLA under this variant.
>
> As shown in the table below, this design setting leads to performance degradation due to unstable conditioning introduced by earlier noise samples. **These results empirically confirm that retaining the entire diffusion chain introduces unnecessary variance and reduces overall model robustness**.
>
> Method     | Without Earlier Noise | With Earlier Noise
> -----------|------------------------|---------------------
> HybridVLA  | 0.74                   | 0.65
>
> ---
> ## **[W2]. Different from previous works**
>
> Thank you for your detailed comments. In the robotics domain, the joint training of diffusion and autoregressive action generation in our HybridVLA is non-trivial. We carefully design a robotics-specific token sequence to model the contextual information essential for robotic manipulation and make the first attempt to demonstrate that jointly optimizing both generation branches can mutually enhance each other, leading to more robust action generation.
>
> ### A. Difference with unified VLMs:
>
>  **Unlike** prior works [69,70] that focus on unifying image and text generation, HybridVLA targets a fundamentally different problem that collaborative model continuous SE(3) action for robotic control. Our contribution lies not just in BOD/EOD tokens, but in several key components:
>
> (1) **A robotics-specific token sequence formulation is designed to embed diffusion modeling into the LLM’s token sequence, leveraging its pretrained knowledge.** This design allows the model to integrate the Markovian denoising steps of diffusion into the LLM’s next-token prediction process. Such integration enables diffusion-based action generation to benefit from the pretrained LLM backbone, treating each denoising step as a reasoning iteration analogous to those in LLMs. Additionally, the diffusion action tokens can provide explicit continuous action latent conditioning for the subsequent autoregressive generation.
>
> (2) **A new approach to jointly optimizing unified action representations using diffusion and autoregressive generation.** In our HybridVLA, both generative branches aim to approximate the same conditional action distribution, thereby jointly generating distributional representations that align with the action distribution under a shared LLM backbone.  This approach differs from general VLMs that generate image and text modalities separately, as it facilitates mutually reinforcing optimization and promotes more robust action representation learning.
>
> (3)  **A collaborative action ensemble mechanism that adaptively fuses both generation branches at test time.** This has not been explored in related work. Specifically, we empirically and innovatively observe that combining the two generation branches not only enables mutual enhancement but also reveals that each branch excels in different tasks or trajectories. As a result, this ensemble mechanism provides a simple yet effective output strategy that leads to more stable control in both simulation and real-world robotic environments.
>
>
>
> ### B. Difference with diffusion-based VLA methods:
>
> Recent diffusion-based VLA methods in the robotics domain, such as Pi0 [13] and CogAct [14], typically attach a separate diffusion head that is trained from scratch after a VLM. These methods use the multimodal latent features extracted by the VLM as static conditions. Meanwhile, existing diffusion-based VLA approaches have not explored the joint optimization of action representations using both diffusion and autoregressive generation. In contrast, HybridVLA is the first to integrate diffusion and autoregressive generation within a unified LLM backbone for collaborative action modeling. It treats each denoising step as a reasoning iteration and designs a robotics-specific unified token sequence to inherit the pretrained reasoning paradigm of the VLM.
>
> ---
> **Finally, we sincerely thank the reviewer for the thoughtful comments and valuable time. We hope our detailed responses address your concerns. If any issues remain, please let us know, and we will respond promptly.**

---

> ### Author Response · Authors · 2025-08-04
> **Further discussion to Reviewer RfcC**
>
> Dear Reviewer RfcC,
>
> First of all, thank you very much for taking the time to review our paper and for your valuable comments. We have taken your concerns seriously and provided detailed responses. Specifically, we clarified that HybridVLA does not feed the entire diffusion chain as tokens into the LLM. Instead, only a single noisy sample from the current denoising step is iteratively fed into the LLM to predict the corresponding noise. We sincerely apologize for any confusion caused, and we will revise Figures 1 and 2, as well as the corresponding description of the diffusion token modeling, in the final version. In addition, we have provided empirical evaluations to support the effectiveness of both our diffusion token modeling approach and the perspective you raised.
>
> As the rebuttal period has now passed the halfway point, we sincerely hope our responses have addressed your concerns. If you find our clarifications satisfactory, we would greatly appreciate your consideration in updating the score, as it would be a meaningful acknowledgment of our efforts. Should you have any further questions or suggestions, we would be happy to discuss them.
>
> Paper 16855 authors

---

> > ### Comment · Reviewer_RfcC · 2025-08-05
> >
> > Thank you for your detailed response. You have addressed my major concerns, and I will revise my score accordingly

---

### Note · Authors · 2025-08-12

Dear Area Chair and Reviewers,

### **1. Strengths Highlighted by Reviewers**
We sincerely thank you for your valuable feedback, which strengthened our paper, and are honored that its novelty and intuition were widely recognized. Reviewer RfcC highlighted that **our core idea “makes sense and is well-motivated,”** Reviewer cjex commended that our paper **“seamlessly incorporates diffusion denoising into the next-token prediction process,”** and Reviewer Ez2J affirmed that our **HybridVLA framework is “novel and also important for the whole community.”**

### **2. Our Rebuttal Responses**
Through constructive discussion, we clarified key technical aspects of our diffusion chain modeling to RfcC. We appreciate that our responses **addressed your major concerns, resulting in a revised positive score.** Following cjex’s comments, we conducted additional evaluations on contextual reasoning, recent VLA comparisons, and simulation generalization, and we are honored to have received **positive recognition for addressing these concerns.** For Ez2J, we validated the intuition of integrating diffusion and autoregressive action generation. Subsequently, our work was recognized for **“achieving the goal of a generalist model: ensuring semantic generalization while retaining accurate action execution.”**

Due to time limits in the discussion, we hope our multiple rounds of responses have addressed Reviewer emK6's concerns. First, we apologize if our writing inadvertently conveyed unintended viewpoints, contributing to emK6's main concern. Revisions will be made accordingly to enhance the rigor in describing existing work limitations. Meanwhile, we will incorporate contextual reasoning–related experiments (acknowledged by cjex and Ez2J), PCA distribution analysis (acknowledged by Ez2J), and two additional experiments with corresponding clarifications from the rebuttal to further support our motivation and innovation.

### **3. Closing Statement**
Finally, we are confident that HybridVLA marks a key step toward more robust VLA models. It is the first VLA framework to integrate diffusion and autoregressive action generation with a unified LLM backbone. We validated that the two components in this collaborative generation can mutually enhance each other through quantitative experiments and feature distribution analysis. HybridVLA achieves SOTA performance in both simulation and real-world settings, while demonstrating strong generalization to unseen configurations.

---

### Decision · Program_Chairs · 2025-09-17

**Decision:**

Reject

**Comment:**

This paper introduces HybridVLA that learns a single VLA that is trained with both autoregressive and diffusion objectives. While the rebuttal addressed many concerns initially raised by the reviewers, the paper still has a mixed set of reviews (2, 4, 4, 5). In particular, the most negative Reviewer emK6 raised a point that this paper lacks a clear motivation and technical understanding on why and how integrating both objectives creates a synergy. This point is also similarly raised by Reviewers Ez2J and cjex but their concerns are resolved in the rebuttal phase.

After carefully reading and reviewing the paper, private messages to AC, and the discussion between authors and reviewers, I ended up partially agreeing with the Reviewer emK6’s point. I don’t think every paper necessarily needs to have a theoretical understanding of everything, but I agree that the paper lacks clarity in its motivation and explaining why using both objectives could be helpful. It seems to me that the paper can be much improved by providing clear writing or investigation on *why* and *how* diffusion-based and autoregressive-based VLAs are different, and *why* and *how* incorporating both objectives can be helpful. I would like to note that the authors’ rebuttal partially addressed these points during the discussion phase by providing additional generalization experiments, but its current version is not ready for NeurIPS 2025 from the perspective of the area chair. Thus I vote to reject the paper.